# Modular metabolite assembly in *Caenorhabditis elegans* depends on carboxylesterases and formation of lysosome-related organelles

Henry H Le[1†], Chester JJ Wrobel[1†], Sarah M Cohen[2], Jingfang Yu[1], Heenam Park[2], Maximilian J Helf[1], Brian J Curtis[1], Joseph C Kruempel[3], Pedro Reis Rodrigues[1], Patrick J Hu[4], Paul W Sternberg[2]*, Frank C Schroeder[1]*

[1]Boyce Thompson Institute and Department of Chemistry and Chemical Biology, Cornell University, Ithaca, United States; [2]Division of Biology and Biological Engineering, California Institute of Technology, Pasadena, United States; [3]Department of Molecular and Integrative Physiology, University of Michigan Medical School, Ann Arbor, United States; [4]Departments of Medicine and Cell and Developmental Biology, Vanderbilt University School of Medicine, Nashville, United States

*For correspondence:
pws@caltech.edu (PWS);
fs31@cornell.edu (FCS)

†These authors contributed equally to this work

Competing interests: The authors declare that no competing interests exist.

**Abstract** Signaling molecules derived from attachment of diverse metabolic building blocks to ascarosides play a central role in the life history of *C. elegans* and other nematodes; however, many aspects of their biogenesis remain unclear. Using comparative metabolomics, we show that a pathway mediating formation of intestinal lysosome-related organelles (LROs) is required for biosynthesis of most modular ascarosides as well as previously undescribed modular glucosides. Similar to modular ascarosides, the modular glucosides are derived from highly selective assembly of moieties from nucleoside, amino acid, neurotransmitter, and lipid metabolism, suggesting that modular glucosides, like the ascarosides, may serve signaling functions. We further show that carboxylesterases that localize to intestinal organelles are required for the assembly of both modular ascarosides and glucosides via ester and amide linkages. Further exploration of LRO function and carboxylesterase homologs in *C. elegans* and other animals may reveal additional new compound families and signaling paradigms.

## Introduction

Recent studies indicate that the metabolomes of animals, from model systems such as *Caenorhabditis elegans* and Drosophila to humans, may include >100,000 of compounds (*da Silva et al., 2015*; *Artyukhin et al., 2018*). The structures and functions of most of these small molecules have not been identified, representing a largely untapped reservoir of chemical diversity and bioactivities. In *C. elegans* (*Girard et al., 2007*), a large modular library of small-molecule signals, the ascarosides, are involved in almost every aspect of its life history, including aging, development, and behavior (*Schroeder, 2015*; *Butcher, 2017*; *Butcher et al., 2007*; *Jeong et al., 2005*). The ascarosides represent a structurally diverse chemical language, derived from glycosides of the dideoxysugar ascarylose and hydroxylated short-chain fatty acid (*Figure 1a*; *von Reuss et al., 2012*). Structural and functional specificity arises from optional attachment of additional moieties to the sugar, for example indole-3-carboxylic acid (e.g. icas#3 (1)), or carboxy-terminal additions to the fatty acid chain, such as *p*-aminobenzoic acid (PABA, as in ascr#8 (2)) or *O*-glucosyl uric acid (e.g. uglas#11 (3), *Figure 1b*; *Artyukhin et al., 2018*; *Artyukhin et al., 2013*; *Bose et al., 2014*; *Aprison and*

*Ruvinsky, 2017*; *Curtis et al., 2020*; *Pungaliya et al., 2009*). Given that even small changes in the chemical structures of the ascarosides often result in starkly altered biological function, ascaroside biosynthesis appears to correspond to a carefully regulated encoding process in which biological state is translated into chemical structures (*Panda et al., 2017*). Thus, the biosynthesis of ascarosides and other *C. elegans* signaling molecules (e.g. nacq#1) (*Ludewig et al., 2019*) represents a fascinating model system for the endogenous regulation of inter-organismal small-molecule signaling in metazoans. However, for most of the >200 recently identified *C. elegans* metabolites (*Artyukhin et al., 2018*; *von Reuss et al., 2012*; *Artyukhin et al., 2013*), biosynthetic knowledge is sparse. Previous studies have demonstrated that conserved metabolic pathways, for example peroxisomal β-oxidation (*Artyukhin et al., 2013*; *Bose et al., 2014*) and amino acid catabolism (*von Reuss et al., 2012*; *Srinivasan et al., 2012*; *Figure 1a*), contribute to ascaroside biosynthesis; however, many aspects of the mechanisms underlying assembly of multi-modular metabolites remains unclear.

Recently, metabolomic analysis of mutants of the Rab-GTPase *glo-1*, which lack a specific type of lysosome-related organelles (LROs, also referred to as autofluorescent gut granules), revealed complete loss of 4′-modified ascarosides (*Panda et al., 2017*). The *glo-1*-dependent LROs are acidic, pigmented compartments that are related to mammalian melanosomes and drosophila eye pigment organelles (*Coburn and Gems, 2013*; *Hermann et al., 2005*). LROs form when lysosomes fuse with other cellular compartments, for example peroxisomes, and appear to play an important role for recycling proteins and metabolites (*Coburn and Gems, 2013*). Additionally, it has been suggested that LROs may be involved in the production and secretion of diverse signaling molecules (*Dell'Angelica et al., 2000*; *Luzio et al., 2014*), and the observation that *glo-1* mutant worms are deficient in 4′-modified ascarosides suggested that intestinal organelles may serve as hubs for their assembly (*Figure 1a*; *Panda et al., 2017*). In addition to the autofluorescent LROs, several other types of intestinal granules have been characterized in *C. elegans*, including lipid droplets (*Cao et al., 2019*) and lysosome-related organelles that are not *glo-1*-dependent (*Tanji et al., 2016*).

Parallel studies of other *Caenorhabditis* species (*Dong et al., 2016*; *Bergame et al., 2019*; *Dolke et al., 2019*) and *Pristionchus pacificus* (*Falcke et al., 2018*), a nematode species being developed as a satellite model system to *C. elegans* (*Rae et al., 2008*), revealed that production of modular ascarosides is widely conserved among nematodes. Leveraging the high genomic diversity of sequenced *P. pacificus* isolates, genome-wide association studies coupled to metabolomic analysis revealed that *uar-1*, a carboxylesterase from the α/ß-hydrolase superfamily with homology to cholinesterases (AChEs), is required for 4′-attachment of an ureidoisobutyryl moiety to a subset of ascarosides, e.g. ubas#3 (4, *Figure 1c*; *Falcke et al., 2018*). Homology searches revealed a large expansion of carboxylesterase (*cest*) homologs in *P. pacificus* as well as *C. elegans* (*Figure 1—figure supplement 1*), and recently it was shown that in *C. elegans*, the *uar-1* homologs *cest-3*, *cest-8*, and *cest-9.2* are involved in the 4′-attachment of other acyl groups in modular ascarosides (*Faghih et al., 2020*). Based on these findings, we posited that *cest* homologs localize to *glo-1*-dependent intestinal granules where they control assembly of modular ascarosides, and perhaps other modular metabolites. In this work, we present a comprehensive assessment of the impact of *glo-1*-deletion on the *C. elegans* metabolome and uncover the central role of *cest* homologs that localize to intestinal granules in the biosynthesis of diverse modular metabolites.

## Results

### Novel classes of LRO-dependent metabolites

To gain a comprehensive overview of the role of *glo-1* in *C. elegans* metabolism, we employed a fully untargeted comparison of the metabolomes of a *glo-1* null mutant and wild-type worms (*Figure 1d*). HPLC–high-resolution mass spectrometry (HPLC–HRMS) data for the *exo*-metabolomes (excreted compounds) and *endo*-metabolomes (compounds extractable from the worm bodies) of the two strains were analyzed using the Metaboseek comparative metabolomics platform, which integrates the *xcms* package (*Tautenhahn et al., 2008*). These comparative analyses revealed that the *glo-1* mutation has a dramatic impact on *C. elegans* metabolism. For example, in negative ionization mode, we detected >1000 molecular features that were at least 10-fold less abundant in the *glo-1 exo-* and *endo*-metabolomes, as well as >3000 molecular features that are 10-fold upregulated

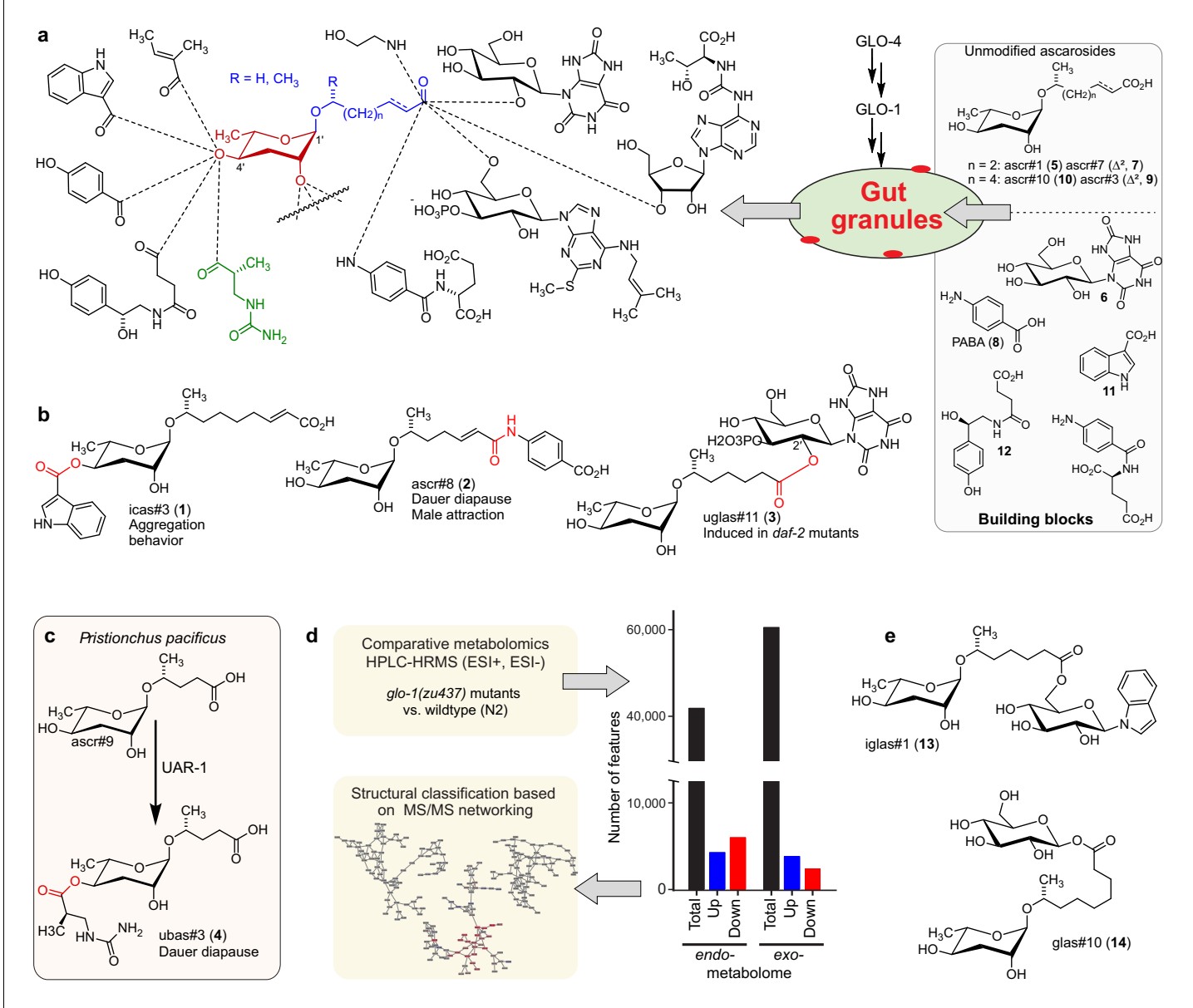

**Figure 1.** Modular ascarosides in nematodes and proposed role of the Rab-GTPase GLO-1. (**a**) Modular ascarosides are assembled from simple ascarosides, e.g. ascr#1 (**5**) or ascr#3 (**9**), and building blocks from other metabolic pathways, e.g. glucosyl uric acid (**6**), *p*-aminobenzoic acid (PABA, **8**) indole-3-carboxylic acid (**11**), or succinyl octopamine (**12**). We hypothesize that *glo-1*-dependent gut granules play a central role in their biosynthesis. (**b**) Examples for modular ascarosides and their biological context. (**c**) UAR-1 in *P. pacificus* converts simple ascarosides into the 4'-ureidoisobutyric-acid-bearing ascarosides, for example ubas#3 (**4**). (**d**) Strategy for comparative metabolomic analysis of LRO-deficient *glo-1* mutants. (**e**) Example for modular ascarosides whose production is increased in *glo-1* mutants.

The online version of this article includes the following source data and figure supplement(s) for figure 1:

**Source data 1.** Source data for *Figure 1d*.
**Figure supplement 1.** Dendrogram of serine hydrolase annotated in *C. elegans* and *Ppa-uar-1* (marked blue).
**Figure supplement 2.** MS peak areas relative to wildtype (N2) of several building blocks of modular ascarosides.
**Figure supplement 2—source data 1.** Source data for *Figure 1—figure supplement 2*.

in *glo-1* mutants. For further characterization of differential features, we employed tandem mass spectrometry (MS$^2$) based molecular networking, a method which groups metabolites based on shared fragmentation patterns (*Figure 1d*, *Figure 2—figure supplements 1–4*; *Wang et al., 2016*). The resulting four MS$^2$ networks – for data obtained in positive and negative ionization mode for the *exo-* and *endo*-metabolomes – revealed several large clusters of features whose abundances were largely abolished or greatly increased in *glo-1* worms. Notably, although some differential MS$^2$ clusters represented known compounds, for example ascarosides, the majority of clusters were found to represent previously undescribed metabolite families.

In agreement with previous studies (*Panda et al., 2017*), biosynthesis of most modular ascarosides was abolished or substantially reduced in *glo-1* mutants, including all 4′-modified ascarosides, e.g. icas#3 (**1**) (*Figure 1b*, *Figure 2—figure supplement 5a*). Similarly, production of ascarosides modified at the carboxy terminus, e.g. uglas#11 (**3**) derived from ester formation between ascr#1 (**5**) and uric acid glucoside (*Curtis et al., 2020*) (**6**), and ascr#8 (**2**), derived from formation of an amide bond between ascr#7 (**7**) and of *p*-amino benzoic acid (**8**), was largely abolished in *glo-1* mutants (*Figure 1a–b*, *Figure 2—figure supplement 5a*). Metabolites plausibly representing building blocks of these modular ascarosides were not strongly perturbed in *glo-1* mutants (*Figure 1—figure supplement 2*). For example, abundances of unmodified ascarosides, for example ascr#3 (**9**) and ascr#10 (**10**), or metabolites representing 4′-modifications, for example indole-3-carboxylic acid (**11**) and octopamine succinate (**12**), were not significantly perturbed in the mutant (*Figure 1a*, *Figure 2—figure supplement 5a*, *Figure 1—figure supplement 2*). In contrast, a subset of modular ascaroside glucose esters (e.g. iglas#1 (**13**) and glas#10 (**14**), *Figure 1e*), was strongly increased in *glo-1* mutants (*Figure 2—figure supplement 5b*). These results suggest that *glo-1*-dependent intestinal organelles function as a central hub for the biosynthesis of most modular ascarosides, with the exception of a subset of ascarosylated glucosides, whose increased production in *glo-1* mutants may be indicative of a shunt pathway for ascarosyl-CoA derivatives (*Zhang et al., 2015*; *Zhang et al., 2016*; *Zhang et al., 2018*), which represent plausible precursors for modular ascarosides modified at the carboxy terminus.

Next, we analyzed the most prominent MS$^2$ clusters representing previously uncharacterized metabolites whose production is abolished or strongly reduced in *glo-1* mutants (*Figure 2*). Detailed analysis of their MS$^2$ spectra indicated that they may represent a large family of modular hexose derivatives incorporating moieties from diverse primary metabolic pathways. For example, MS$^2$ spectra from clusters **I**, **II**, and **III** of the positive-ionization network suggested phosphorylated hexose glycosides of indole, anthranilic acid, tyramine, or octopamine, which are further decorated with a wide variety of fatty acyl moieties derived from fatty acid or amino acid metabolism, for example nicotinic acid, pyrrolic acid, or tiglic acid (*Figure 2*, *Appendix 1—table 1*; *Coburn and Gems, 2013*; *Stupp et al., 2013*). Given the previous identification of the glucosides iglu#1/2 (15/16, *Figure 2e*) and angl#1/2 (17/18), we hypothesized that clusters I, II, and III represent a modular library of glucosides, in which *N*-glucosylated indole, anthranilic acid, tyramine, or octopamine (*O'Donnell et al., 2020*) serve as scaffolds for attachment of diverse building blocks. To further support these structural assignments, a series of modular metabolites based on *N*-glucosylated indole ('iglu') were selected for total synthesis. Synthetic standards for the non-phosphorylated parent compounds of iglu#4 (**19**), iglu#6 (**20**), iglu#8 (**21**), and iglu#10 (**22**) matched HPLC retention times and MS$^2$ spectra of the corresponding natural compounds (*Figure 2—figure supplement 6*), confirming their structures and enabling tentative structural assignments for a large number of additional modular glucosides, including their phosphorylated derivatives, e.g. iglu#12 (**23**), iglu#41 (**24**), angl#4 (cluster II, **25**), and tyglu#4 (cluster III, **26**) (*Figure 2*). The proposed structures include several glucosides of the neurotransmitters tyramine and octopamine, whose incorporation could be verified by comparison with data from a recently described feeding experiment with stable isotope-labeled tyrosine (*O'Donnell et al., 2020*). Similar to ascaroside biosynthesis, the production of modular glucosides is life stage dependent; for example, production of specific tyramine glucosides peaks at the L3 larval stage, whereas production of angl#4 increases until the adult stage (*Figure 2—figure supplement 8*). Notably, modular glucosides were detected primarily as their phosphorylated derivatives, as respective non-phosphorylated species were generally less abundant. In contrast to most ascarosides, the phosphorylated glucosides are more abundant in the *endo*-metabolome than the *exo*-metabolome, suggesting that phosphorylated glucosides may be specifically retained in the body (*Figure 2—figure supplement 7*).

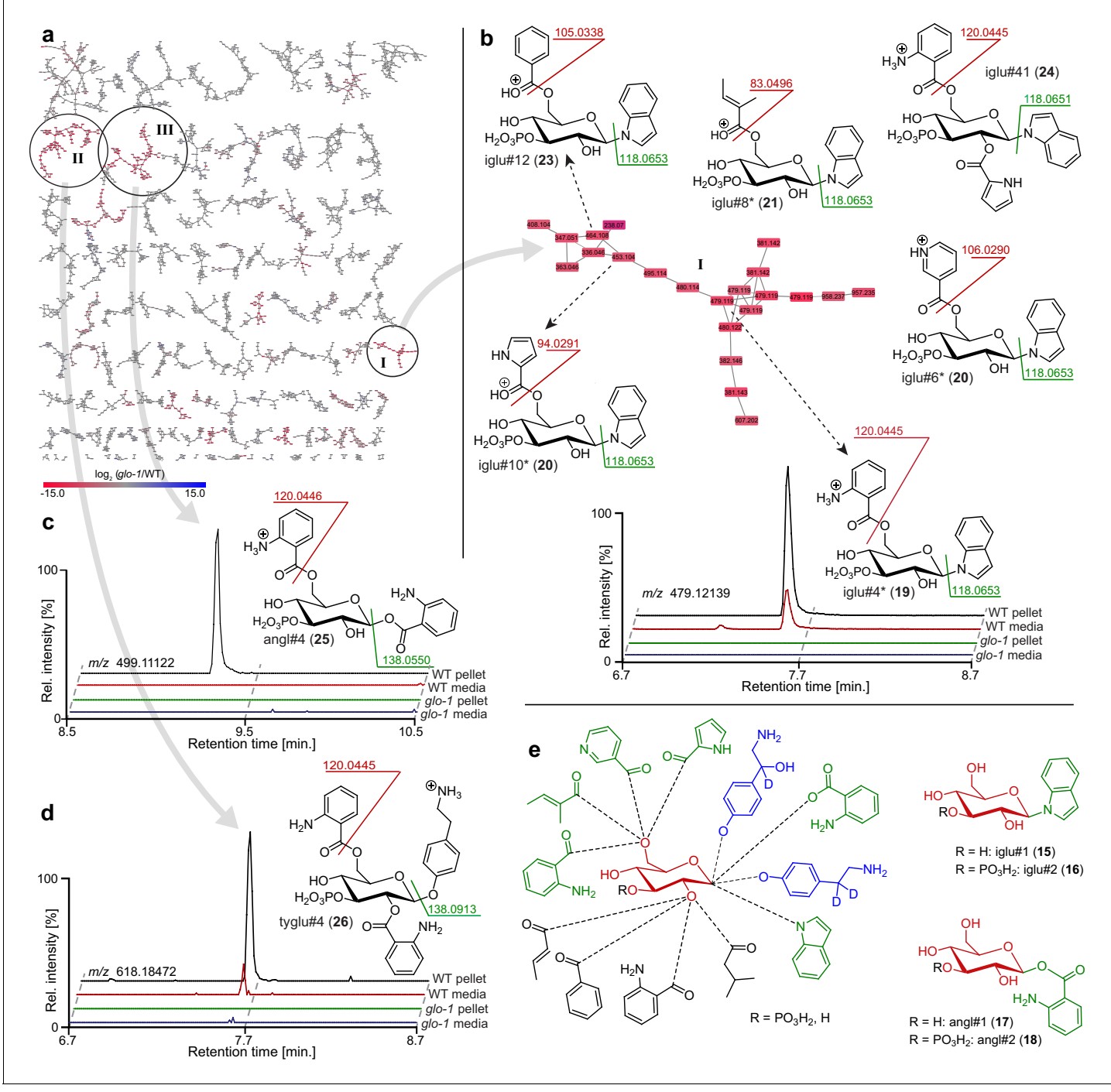

**Figure 2.** Comparative metabolomic analysis of *glo-1* mutants. (**a**) Partial MS² network (positive ion mode) for *C. elegans endo*-metabolome highlighting three clusters of modular glucosides that are down regulated in the *glo-1* mutants (also see *Figure 2—figure supplements 1–4*). Red represents downregulated and blue upregulated features compared to wildtype *C. elegans*. (**b**) Cluster I feature several modular indole glucoside derivatives. Structures were proposed based on MS² fragmentation patterns, also see *Appendix 1—table 1*. Compounds whose non-phosphorylated analogs were synthesized are marked (*). Shown ion chromatograms demonstrate loss of iglu#4 in *glo-1* mutants. (**c,d**) Examples for modular glucosides detected as part of clusters II and III. Ion chromatograms show abolishment of angl#4 (25) (**c**) and tyglu#4 (26) (**d**) production in *glo-1* mutants. (**e**) Modular glucosides are derived from combinatorial assembly of a wide range of building blocks. Incorporation of moieties was confirmed via total synthesis of example compounds (green) or stable isotope labeling (blue). For all compounds, 3-phosphorylation was proposed based on the established structures of iglu#2 (16), angl#2 (18), and uglas#11 (3).

The online version of this article includes the following source data and figure supplement(s) for figure 2:

*Figure 2 continued on next page*

*Figure 2 continued*

**Figure supplement 1.** Full MS$^2$ molecular network of *endo*-metabolome acquired in positive ion mode (left).

**Figure supplement 2.** Full MS$^2$ molecular network of *endo*-metabolome acquired in negative ion mode.

**Figure supplement 3.** Full MS$^2$ molecular network of *exo*-metabolome acquired in positive ion mode.

**Figure supplement 4.** Full MS$^2$ molecular network of *exo*-metabolome acquired in negative ion mode.

**Figure supplement 5.** MS peak areas relative to wildtype (N2) of simple and modular ascarosides, glucosylated ascarosides, and phosphorylated ascarosides in *glo-1* (a, b, c) and *glo-4* (d, e, f) mutant worms.

**Figure supplement 5—source data 1.** Source data for *Figure 2—figure supplement 5a–f*.

**Figure supplement 6.** Identification of iglu metabolites.

**Figure supplement 7.** Concentration of simple and modular glucosides in the *endo*- or *exo*-metabolomes wild-type *C. elegans*.

**Figure supplement 7—source data 1.** Source data for *Figure 2—figure supplement 7*.

**Figure supplement 8.** Production of modular glucosides is life-stage-dependent.

**Figure supplement 8—source data 1.** Source data for *Figure 2—figure supplement 8a–d*.

**Figure supplement 9.** Peak area relative to wildtype (N2) of building blocks of modular glucosides in *glo-1* mutant worms.

**Figure supplement 9—source data 1.** Source data for *Figure 2—figure supplement 9*.

**Figure supplement 10.** Representative ion chromatograms and MS$^2$ spectra of upregulated leucine- and proline-containing peptides.

As in the case of modular ascarosides, the abundances of putative building blocks of the newly identified modular glucosides were not strongly perturbed in *glo-1* mutants. For example, abundances of anthranilic acid, indole, octopamine, and tyramine were not significantly affected in *glo-1* null animals (*Figure 2—figure supplement 9*). Notably, abundances of the glucosides scaffold, e.g. iglu#1 and angl#1, were also largely unaltered or even slightly increased in *glo-1* mutants (*Figure 2—figure supplement 9*). In addition, production of some of the identified modular glucosides, e.g. iglu#5, is reduced but not fully abolished in *glo-1* worms (*Figure 2—figure supplement 6*).

To confirm our results, we additionally compared the *glo-1* metabolome with that of *glo-4* mutants. *glo-4* encodes a predicted guanyl-nucleotide exchange factor acting upstream of *glo-1*, and like *glo-1* mutants, *glo-4* worms do not form LROs (*Hermann et al., 2005*). We found that the *glo-4* metabolome closely resembles that of *glo-1* worms, lacking most of the modular ascarosides and ascarosides detected in wildtype worms (*Figure 2—figure supplement 5c*). Correspondingly, similar sets of compounds are upregulated in *glo-1* and *glo-4* mutants relative to wild type, including ascarosyl glucosides and ascaroside phosphates. Compounds accumulating in *glo-1* and *glo-4* mutant worms further include a diverse array of small peptides (primarily three to six amino acids), consistent with the proposed role of LROs in the breakdown of peptides derived from proteolysis (*Figure 2—figure supplement 10*; *Bird et al., 2009*). Taken together, our results indicate that, in addition to their roles in the degradation of metabolic waste, the LROs serve as hotspots of biosynthetic activity, where building blocks from diverse metabolic pathways are attached to glucoside and ascaroside scaffolds (*Figure 1a*).

## Carboxylesterases are required for modular assembly

Comparing the relative abundances of different members of the identified families of modular glucosides and ascarosides, it appears that combinations of different building blocks and scaffolds are highly specific, suggesting the presence of dedicated biosynthetic pathways. For example, uric acid glucoside, gluric#1 (6), is preferentially combined with an ascaroside bearing a seven-carbon side chain (to form uglas#11, 3), whereas ascarosides bearing a nine-carbon side chain are preferentially attached to the anomeric position of free glucose, as in glas#10 (14) (*Artyukhin et al., 2018*; *von Reuss et al., 2012*). Similarly, tiglic acid is preferentially attached to indole and tyramine glucosides but not to anthranilic acid glucosides (*Appendix 1—table 1*). Given that 4′-modification of ascarosides in *P. pacificus* and *C. elegans* require *cest* homologs, we hypothesized that the biosynthesis of other modular ascarosides as well as the newly identified glucosides may be under the control of *cest* family enzymes (*Falcke et al., 2018*; *Faghih et al., 2020*). From a list of 44 *uar-1* homologs from BLAST analysis (*Appendix 1—table 2*), we selected seven for further study (*Figure 3a*, *Appendix 1—table 3*). The selected homologs are predicted to have intestinal expression, one primary site of small molecule biosynthesis in *C. elegans* (*Artyukhin et al., 2018*), and are closely related to the UAR-1 gene, while representing different sub-branches of the phylogenetic tree. Utilizing a recently optimized CRISPR/Cas9 method, we obtained two null mutant strains for

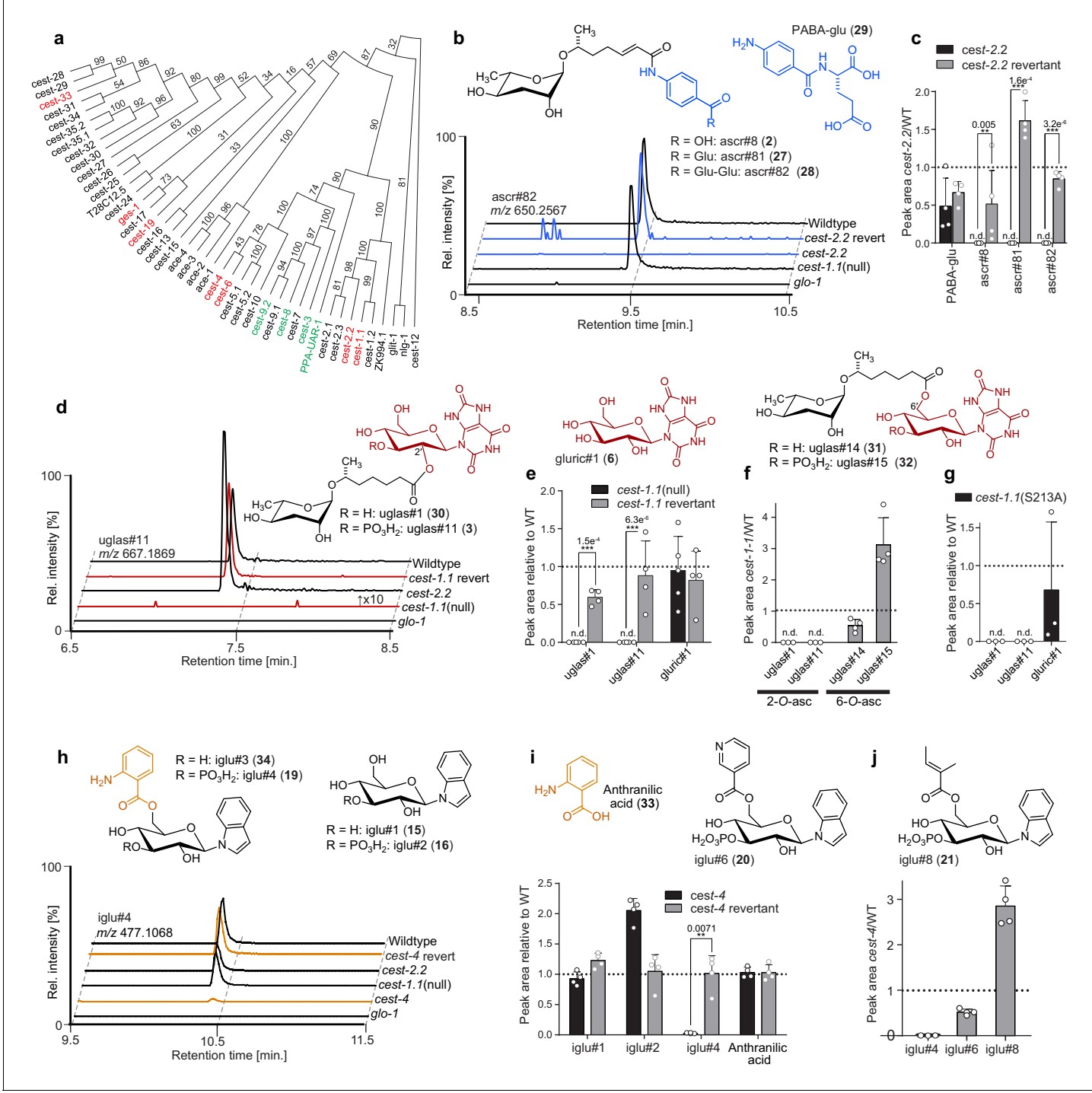

**Figure 3.** Carboxylesterases are required for modular assembly. (**a**) Serine hydrolase dendrogram relating *P. pacificus uar-1* to homologous predicted genes in *C. elegans*. *Ppa-uar-1*, *cest-3*, *cest-8*, *cest-9.2* (green) mediate ester formation at the 4′-position of ascarosides in *P. pacificus* and *C. elegans*. Genes shown in red color were selected for the current study. (**b,c**) Production of ascr8 (**2**), ascr81 (27), and ascr82 (28) is abolished in *cest-2.2* mutants Isogenic revertant strains of the *cest-2.2* null mutants in which the STOP-IN cassette was precisely excised, demonstrate wild-type-like recovery of the associated metabolite. (**d,e**) Production of uglas1 and uglas11 is abolished in *cest-1.1*(null) mutants and recovered in genetic revertants. (**f**) Biosynthesis of positional isomers uglas14 (31) and uglas15 (32) is unaltered or increased in *cest-1.1* mutants (**f**). (**g**) Production of uglas1 and uglas11, but not gluric1, is abolished in *cest-1.1*(S213) mutants. (**h,i**) Production of the anthranilic-acid-modified glucoside iglu4 is largely abolished in *cest-4* mutants and fully recovered in genetic revertants. (**j**) Production of iglu6 (36) and iglu8 (37), whose structures are closely related to that of iglu4, is not abolished in *cest-4* mutants. Ion chromatograms in panels b, d, and g further demonstrate abolishment in *glo-1* mutants. n.d., not detected. Error bars are standard deviation of the mean, and p-values are depicted in the Figure.

*Figure 3 continued on next page*

*Figure 3 continued*

The online version of this article includes the following source data and figure supplement(s) for figure 3:

**Source data 1.** Source data for *Figure 3c,e,f,g,I,j*.
**Figure supplement 1.** Relative abundances of ascr#8 (**2**) and related metabolites in *cest-1.1, cest-2.2, cest-4* mutants, and wild type (N2).
**Figure supplement 1—source data 1.** Source data for *Figure 3—figure supplement 1*.
**Figure supplement 2.** Ion chromatograms demonstrating that abundances of potential precursors of (**a**) *cest-1.1*-dependent, (**b**) *cest-2.2*-dependent, and (**c**) *cest-4*-dependent metabolites is large unchanged in the corresponding mutants.
**Figure supplement 3.** Ion chromatograms demonstrating recovery of (**a**) *cest-1.1*-dependent, (**b**) *cest-8*-dependent, (**c**) *cest-2.2*-dependent, (**d**) *cest-4*-dependent metabolites from CRISPR/Cas9 reversions of the corresponding null mutants.
**Figure supplement 4.** Relative abundance of other indole containing glucosides in *cest-4* mutants, demonstrating that *cest-4* is specifically required for the production of iglu#3 (**34**) and #4 (**19**).
**Figure supplement 4—source data 1.** Source data for *Figure 3—figure supplement 4*.

five of the selected genes (*Wang et al., 2018*). Mutants for the remaining two homologs, *ges-1* and *cest-6*, had been previously obtained (*Appendix 1—table 3*). We then analyzed the *exo-* and *endo-* metabolomes of this set of mutant strains by HPLC-HRMS to identify features that are absent or strongly downregulated in null mutants of a specific candidate gene compared to wildtype worms and all other mutants in this study. We found that two of the seven tested homologs (*cest-1.1, cest-2.2*) are defective in the production of two different families of modular ascarosides, whereas *cest-4* mutants were defective in the biosynthesis of a specific subset of modular indole glucosides (*Figure 3*). The metabolomes of mutants for the remaining four *cest* homologs did not exhibit any significant differences compared to wild type under the tested conditions.

Analysis of the metabolomes of the two *cest-2.2* null mutants revealed loss of dauer pheromone component and male attractant ascr#8 (**2**) as well as of the closely related ascr#81 (**27**) and ascr#82 (**28**) (*Figure 3b*, *Figure 3—figure supplements 1* and *2b*). Biosynthetically, the ascr#8 family of ascarosides are derived from amide formation between ascr#7 (ΔC7, **7**) and folate-derived *p*-amino-benzoic acid (PABA, **8**), PABA-glutamate (**29**), or PABA-diglutamate, respectively. We did not detect any significant reduction in the production of plausible ascr#8 precursors, including PABA and PABA-glutamate, or ascr#7 (*Figure 3b*, *Figure 3—figure supplement 2b*). Biosynthesis of ascr#8, ascr#81, and ascr#82 was recovered in *cest-2.2* mutant worms in which the *cest-2.2* sequence had been restored to wild type using CRISPR/Cas9 (*Figure 3c*, *Figure 3—figure supplement 3b*). These results indicate that CEST-2.2 is required specifically for biosynthesis of the amide linkage between the carboxy terminus of ascr#7 and PABA derivatives, in contrast to the implied functions of UAR-1, CEST-8, CEST-3, and CEST-9.2, which are involved in the formation of ester bonds between various head groups and the 4'-hydroxy group of ascarylose (*Falcke et al., 2018*; *Faghih et al., 2020*).

In *cest-1.1* null mutants (*cest-1.1*(null)), biosynthesis of the nucleoside-like ascaroside uglas#1 (**30**) and its phosphorylated derivative uglas#11 (**3**) was abolished (*Figure 3d*, *Figure 3—figure supplement 2a*). uglas#1 and uglas#11 are derived from the attachment of ascr#1, bearing a seven carbon (C7) side chain, to the uric acid gluconucleoside gluric#1 (**6**). Production of ascr#1 (**5**) and gluric#1 (**6**), representing plausible building blocks of uglas#1 (**30**), was not reduced (*Figure 3—figure supplement 2a*). Furthermore, production of uglas#14 (**31**) and uglas#15 (**32**), isomers of uglas#1 and uglas#11 bearing the ascarosyl moiety at the 6' position instead of the 2' position, was not abolished but rather slightly increased in *cest-1.1*(null) (*Figure 3d–e*). These results indicate that CEST-1.1 is required for the formation of the ester bond specifically between ascr#1 (**5**) and the 2'-hydroxyl group in gluric#1. As in the case of *cest-2.2*, biosynthesis of uglas#1 and uglas#11 was fully recovered in *cest-1.1* mutant worms in which the *cest-1.1* sequence had been restored to wild type using CRISPR/Cas9 (*Figure 3f*, *Figure 3—figure supplement 3a*).

Sequence alignment with human AChE suggested that serine 213 is part of the conserved catalytic serine-histidine-glutamate triad of CEST-1.1 (*Figure 4—figure supplement 1*). To test whether disruption of the catalytic triad would affect production of *cest-1.1*-dependent metabolites, we generated a point mutant, *cest-1.1*(S213A). As in *cest-1.1*(null), production of uglas#1 (**30**) and uglas#11 (**3**) was fully abolished in *cest-1.1*(S213A), whereas production of gluric#1 was not affected (*Figure 3g*).

Previous work implicated *cest-1.1* with longevity phenotypes associated with argonaute-like gene 2 (*alg-2*) (*Aalto et al., 2018*). *alg-2* mutant worms are long lived compared to wild type and their

long lifespan was further shown to require *daf-16*, the sole ortholog of the FOXO family of transcription factors in *C. elegans*, as well as *cest-1.1*. Moreover, uglas#11 biosynthesis is significantly increased in mutants of the insulin receptor homolog *daf-2*, a central regulator of lifespan in *C. elegans* upstream of *daf-16* (*Curtis et al., 2020*). These findings suggest the possibility that the production of uglas ascarosides underlies the *cest-1.1*-dependent extension of adult lifespan in *C. elegans*.

In contrast to our results for *cest-1.1* and *cest-2.2* mutants, comparative metabolomic analysis of the *cest-4* mutant strains did not reveal any defects in the biosynthesis of known ascarosides. Instead, we found that the levels of a specific subset of modular anthranilic acid (33) bearing indole glucosides, including iglu#3 (34) and its phosphorylated derivative iglu#4 (35) were abolished in the *cest-4* mutant worms (*Figure 3h*). Abundances of the putative precursor glucosides, iglu#1 (15) and iglu#2 (16), were not significantly changed in *cest-4* (*Figure 3i*, *Figure 3—figure supplement 2c*). Notably, production of other indole glucosides, e.g. iglu#6 (36) and iglu#8 (37), was not significantly reduced in *cest-4* worms (*Figure 3j*, *Figure 3—figure supplement 4*). Biosynthesis of iglu#3 and iglu#4 was restored to wild-type levels in genetic revertant strains for *cest-4* (*Figure 3i*, *Figure 3—figure supplement 3c*). Therefore, it appears that *cest-4* is specifically required for attachment of anthranilic acid to the 6′ position of glucosyl indole precursors, whereas attachment of tiglic acid, nicotinic acid, and other moieties is *cest-4*-independent (*Figure 3j*, *Figure 3—figure supplement 4*). The role of *cest-4* in the biosynthesis of the iglu family of modular glucosides thus parallels that of *cest-1.1* in the biosynthesis of the uglas ascarosides: whereas *cest-4* appears to be required for the attachment of anthranilic acid (33) to the 6′ position of a range of indole glucosides, *cest-1.1* appears to be required for attaching the ascr#1 side chain to the 2′ position in uric acid glucosides.

## CEST-2.2 localizes to intestinal granules

All *cest* homologs selected for this study exhibit domain architectures typical of the α/ß-hydrolase superfamily of proteins, including a conserved catalytic triad, and further contain a predicted disulfide bridge, as in mammalian AChE (*Soreq and Seidman, 2001*; *Figure 4—figure supplement 1*). The *cest* genes also share homology with neuroligin, a membrane bound member of the α/ß-hydrolase fold family, that mediates the formation and maintenance of synapses between neurons (*Bemben et al., 2015*). Sequence analysis suggests that five of the seven CEST homologs studied here are membrane anchored, given the presence of a predicted *C*-terminal transmembrane domain (*Krogh et al., 2001*) (consisting of ~20 residues), with the *N* terminus on the luminal side of a vesicle or organelle (*Figure 4—figure supplement 2*). Since the production of all so far identified *cest*-dependent metabolites is abolished in *glo-1* mutants, it seemed likely that the CEST proteins localize to intestinal granules. To test this idea, we created a mutant strain that express *cest-2.2* *C*-terminally tagged with mCherry at the native genomic locus to avoid potentially confounding effects of overexpression. The red fluorescent mCherry was chosen because of the strong green autofluorescence of the LROs (*Coburn and Gems, 2013*). We confirmed that production of all *cest-2.2*-dependent metabolites, including ascr#8 (**2**), ascr#81 (**27**), and ascr#82 (**28**) was not significantly altered in *cest-2.2*-mCherry mutants (*Figure 4a*), indicating that CEST-2.2 remained functional. Imaging of wild-type adult worms revealed strong green and weaker red autofluorescence in circular features in intestinal cells, consistent with LROs. In addition, *cest-2.2*-mCherry-tagged worms showed red fluorescence in a distinct set of intestinal granules that showed little if any autofluorescence (*Figure 4b*, *Figure 4—figure supplements 3–4*). It is unclear whether mCherry also localizes to the strongly autofluorescent granules, as we cannot distinguish the mCherry signal from the red component of the autofluorescence, given relatively low CEST-2.2-mCherry expression in this non-overexpressing strain. Taken together, it appears that CEST-2.2-mCherry localizes to a subset of intestinal organelles that is partly distinct from the autofluorescent LROs. Further studies are required to determine if CEST-2.2-mCherry co-localizes with other intestinal granule markers, specifically GLO-1 and the lysosomal marker LMP-1.

## *Glo-1*-dependent metabolites in *C. briggsae*

In addition to *C. elegans* and *P. pacificus*, modular ascarosides have been reported from several other *Caenorhabditis* species (*Dong et al., 2020*; *Kanzaki et al., 2018*), including *C. briggsae* (*Dong et al., 2016*; *von Reuss, 2018*). To assess whether the role of LROs in the biosynthesis of modular metabolites is conserved across species, we created two *Cbr-glo-1* (CBG01912.1) knock-

out strains using CRISPR/Cas9. As in *C. elegans*, *Cbr-glo-1* mutant worms lacked autofluorescent LROs, which are prominently visible in wild-type *C. briggsae* (*Figure 5—figure supplement 1*). Comparative metabolomic analysis of the *endo-* and *exo-*metabolomes of wild-type *C. briggsae* and the *Cbr-glo-1* mutant strains revealed that biosynthesis of all known modular ascarosides is abolished in *Cbr-glo-1* worms, including the indole carboxy derivatives icas#2 (35) and icas#6.2 (36), which are highly abundant in wild-type *C. briggsae* (*Figure 5a*; *Dong et al., 2016*). In addition, the *C. briggsae* MS[2] networks included several large *Cbr-glo-1*-dependent clusters representing modular glucosides,

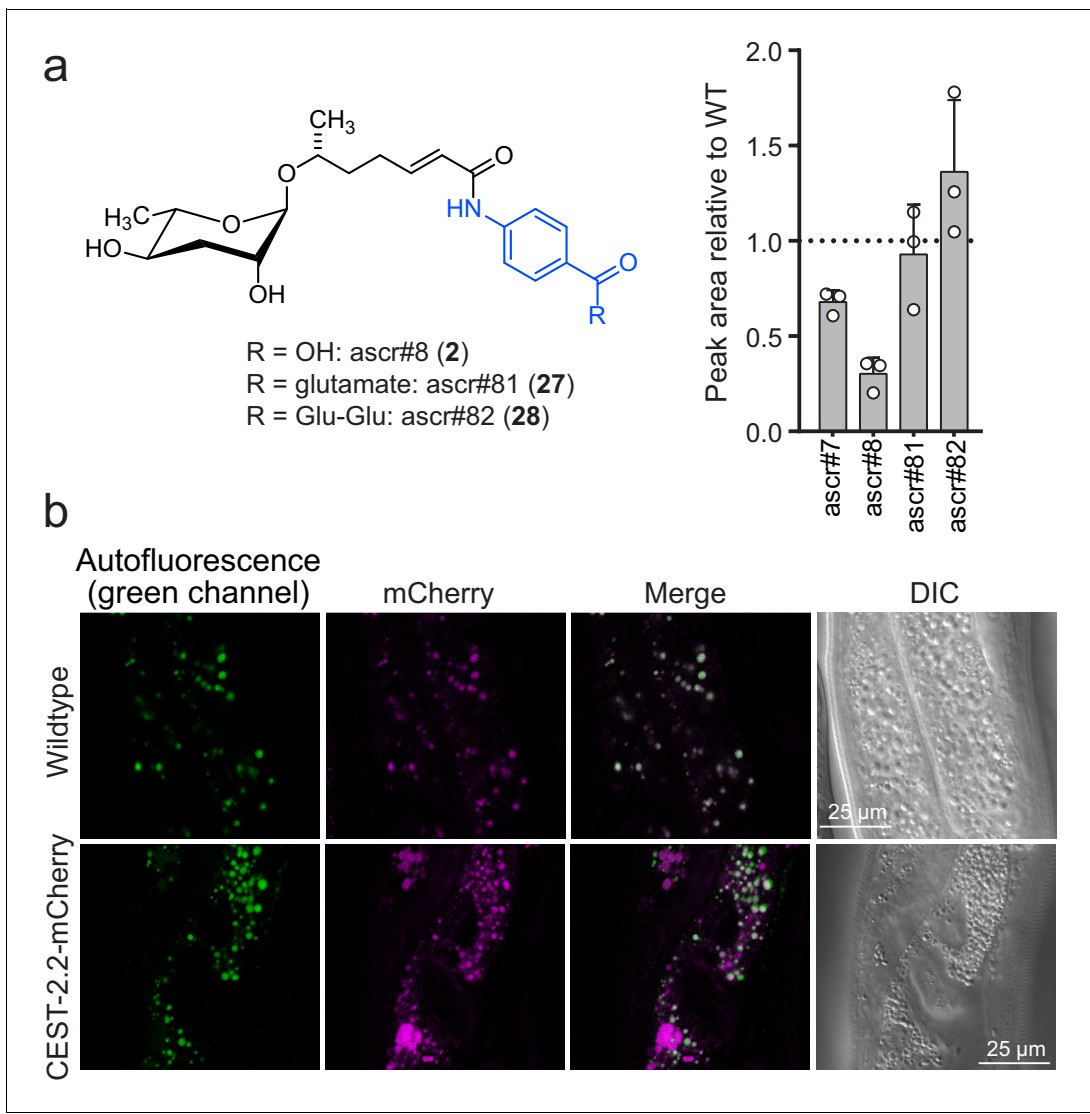

**a**

R = OH: ascr#8 (**2**)
R = glutamate: ascr#81 (**27**)
R = Glu-Glu: ascr#82 (**28**)

**b**

Autofluorescence (green channel) | mCherry | Merge | DIC

Wildtype

CEST-2.2-mCherry

**Figure 4.** CEST-2.2 localizes to intestinal granules. (**a**) Relative amounts of *cest-2.2*-dependent metabolites in worms expressing *C*-terminally mCherry-tagged CEST-2.2. (**b**) Red fluorescence in intestinal granules in wild-type and *cest-2.2*-mCherry gravid adults. Top, wild-type (N2) control; bottom, *cest-2.2*-mCherry worms.
The online version of this article includes the following source data and figure supplement(s) for figure 4:

**Source data 1.** Source data for *Figure 4a*.
**Figure supplement 1.** Amino acid sequence alignments of human acetyl cholinesterase (hAChE), *P. pacificus* UAR-1, and *C. elegans* CEST-1.1, CEST-2.2, and CEST-4.
**Figure supplement 2.** Transmembrane domain prediction for CEST proteins in this study (*cest-1.1, cest-2.2, cest-4, cest-6, cest-19, cest-33, ges-1*).
**Figure supplement 3.** Red fluorescence in intestinal granules in gravid adults, expressing *C*-terminally mCherry-tagged CEST-2.2.
**Figure supplement 4.** Co-localization of green and red autofluorescence in wild-type (N2) gravid adults.

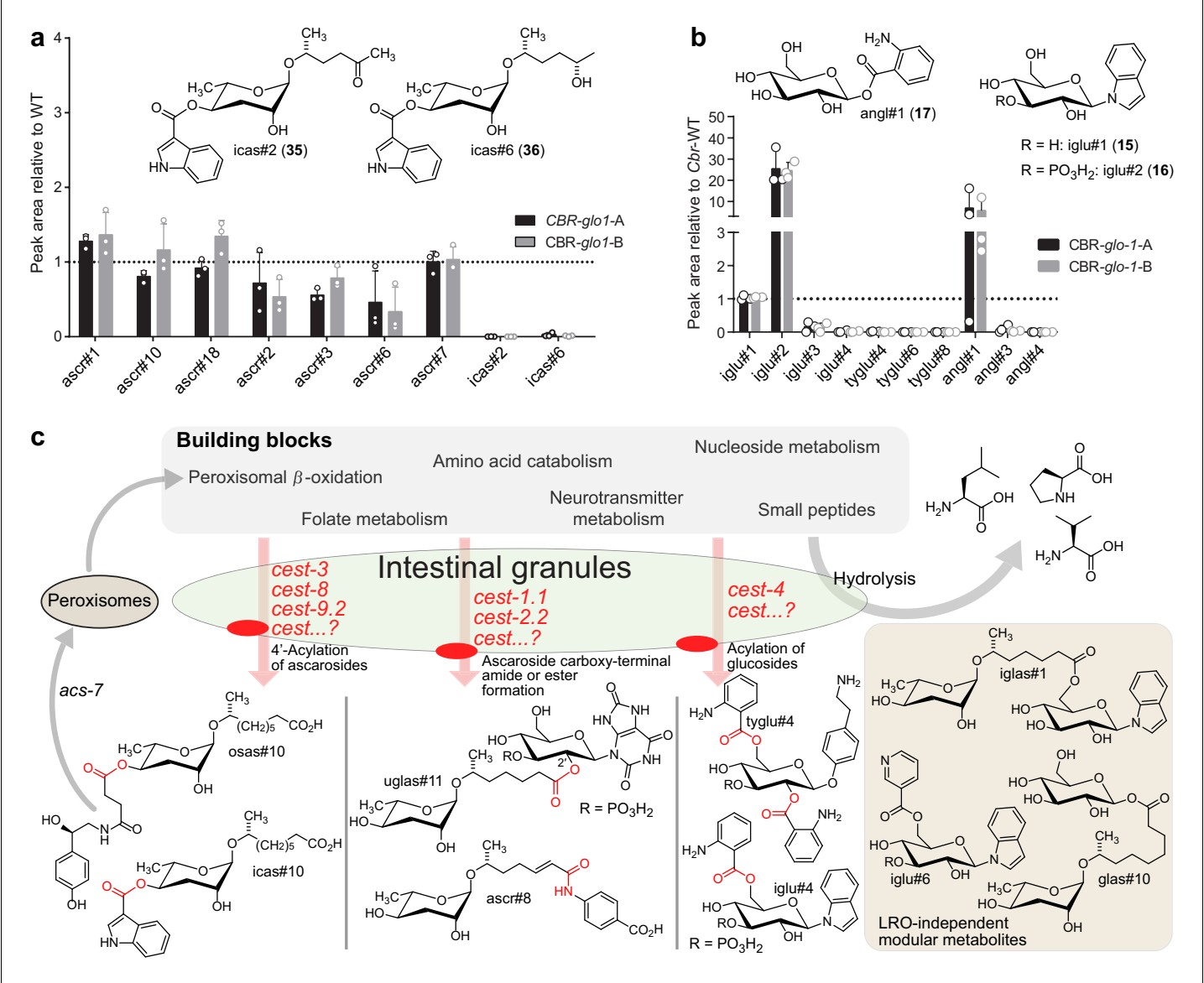

**Figure 5.** Relative abundance of (a) simple and modular ascarosides and (b) simple and modular glucosides in the *endo*-metabolome of *Cbr-glo-1* mutants relative to wild-type *C. briggsae*. n.d., not detected. (c) Model for modular metabolite assembly. CEST proteins (membrane-bound in the LROs, red) mediate attachment of building blocks from diverse metabolic pathways to glucose scaffolds and peroxisomal β-oxidation-derived ascarosides via ester and amide bonds. Some of the resulting modular ascarosides may undergo additional peroxisomal β-oxidation following activation by *acs-7* (**Dolke et al., 2019**).

The online version of this article includes the following source data and figure supplement(s) for figure 5:

**Source data 1.** Source data for *Figure 5a–b*.
**Figure supplement 1.** Gut granules in *C. briggsae*.

including many of the compounds also detected in *C. elegans*, for example iglu#4 and angl#4. As in *C. elegans*, production of unmodified glucoside scaffolds, e.g. iglu#1 (15) and angl#1 (17), was not reduced or increased in *Cbr-glo-1* mutants, whereas biosynthesis of most modular glucosides derived from attachment of additional moieties to these scaffolds was abolished (*Figure 5b*). Taken together, these results indicate that the role of LROs as a central hub for the assembly of diverse small molecule architectures, including modular glucosides and ascarosides, may be widely conserved among nematodes (*Figure 5c*).

## Discussion

Our results indicate that in *C. elegans* the Rab-GTPase *glo-1,* which is required for formation of intestinal LROs, plays a central role in the biosynthesis of several large compound families derived from modular assembly via *cest* homologs. Formation of the autofluorescent LROs via *glo-1* is reminiscent of the roles of its human orthologs RAB32 and RAB38, which are required for the formation of melanosomes, and perhaps other LROs (*Wasmeier et al., 2006*; *Marks et al., 2013*). Lysosomes and LROs are generally presumed to function in autophagy, phagocytosis, and the hydrolytic degradation of proteins, and Rab32 family GTPases have been shown to be required for these processes in diverse organisms (*Morris et al., 2018*). Consistent with the notion that lysosomes and LROs are degradation hotspots, many of the building blocks of the identified modular ascarosides and glucosides are derived from catabolic pathways, for example, anthranilic acid is derived from tryptophan catabolism, uric acid stems from purine metabolism, and the short chain ascarosides are the end products of peroxisomal β-oxidation of very long-chain precursors. Importantly, although our results indicate that carboxylesterases participate in *glo-1*-dependent modular metabolite assembly, additional studies are required to clarify whether the intestinal compartments that carboxylesterases localize to also contain GLO-1 and the lysosomal marker LMP-1, as is the case for the autofluorescent LROs (*Tanji et al., 2016*).

Further, our results demonstrate that the modular assembly paradigm extends beyond ascarosides. The modular glucosides represent a previously unknown family of nematode metabolites. In contrast to the well-established role of modular ascarosides as pheromones, it is unknown whether modular glycosides serve specific biological functions, for example as signaling molecules; however, their specific biosynthesis via *cest-4* as well as their life-stage-dependent production strongly supports this hypothesis (*Figure 2—figure supplement 8*). Like the ascaroside pheromones, some modular glucosides are excreted into the media, suggesting that they could be involved in interorganismal communication. Identifying developmental and environmental conditions that affect modular glucoside production, as well as a more comprehensive understanding of their biosyntheses, may help uncover potential signaling and other biological roles. In particular, the apparent peroxisomal origin of the ascaroside scaffolds suggests a link between peroxisome and gut granule activity, perhaps via pexophagy (*Sakai et al., 2006*), and characterization of the role of autophagy for gut granule-dependent metabolism may contribute to uncovering the functions of modular glucoside and ascarosides. A connection to autophagy is also suggested by our previous finding (*Panda et al., 2017*) that production of modular ascarosides is reduced in mutants of *atg-18* (*Palmisano and Meléndez, 2019*), which is essential for autophagy.

The high degree of selectivity in which different building blocks are combined in the modular ascarosides and glucosides strongly suggests that these compounds, despite their numbers and diversity, represent products of dedicated enzymatic pathways, as has recently been established for 4′-acylated ascarosides. Our results revealed a wider range of biosynthetic functions associated with *cest* homologs, including esterification and amide formation at the carboxy terminus of ascarosides and acylation of glucosides (*Figure 5c*). Notably, all *cest* null mutants whose metabolomes have been characterized so far are defective in the biosynthesis of one or a few compounds sharing a specific structural feature, further supporting the view that these selectively assembled molecular architectures serve dedicated functions.

All CEST proteins that so far have been associated with modular metabolite assembly contain membrane-anchors and exhibit domain architectures typical of serine hydrolases of the AChE family, including an α/β-hydrolase fold, a conserved catalytic serine-histidine-glutamate triad, and bridging disulfide cysteines (*Figure 4—figure supplement 1*; *Soreq and Seidman, 2001*). While our efforts at heterologous expression of CEST proteins were unsuccessful, the finding that mutation of the catalytic serine in *cest-1.1*(S213A) abolished production of all *cest-1.1*-dependent compounds suggests that CEST enzymes directly participate in the biosynthesis of modular metabolites. Therefore, we hypothesize that CEST proteins, after translating from the endomembrane system to *glo-1*-dependent intestinal organelles, partake in the assembly of diverse ascaroside or glucoside-based architectures via acyl transfer from corresponding activated intermediates, e.g. CoA or phosphate esters (*Soreq and Seidman, 2001*; *Vaz and Wanders, 2002*). α/β-hydrolase fold enzymes are functionally highly diverse (*Rauwerdink and Kazlauskas, 2015*) and include esterases, peptidases, oxidoreductases, and lyases, serving diverse biosynthetic roles in animals, plants (*Mindrebo et al., 2016*), and

bacteria (*Zheng et al., 2016*). While acyltransferase activity is often observed as a side reaction for esterases and lipases, α/β-hydrolase fold enzymes can function as dedicated acyltransferases, for example in microbial natural product biosyntheses (*Rauwerdink and Kazlauskas, 2015*; *Lejon et al., 2008*). Additional biochemical studies will be required to delineate the exact mechanisms by which *cest* homologs contribute to modular metabolite assembly in nematodes.

Finally, although our results indicate that *glo-1* is required for the biosynthesis of most modular metabolites we have detected so far, it is notable that some modular ascarosides, e.g. iglas#1 (13), and modular glucosides, e.g. iglu#6 (20) and iglu#8 (21), do not appear to be *glo-1*-dependent (*Figure 2—figure supplement 7*). This suggests that diverse cell compartments contribute to modular metabolite biosynthesis and may also indicate that not all CEST proteins are delivered to the same cellular compartment. Similarly, *glo-1* mutants continue to generate the simple glucosides and ascarosides that serve as scaffolds for further elaboration via CEST proteins, which may be derived from UDP-glycosyltransferases (*Mackenzie et al., 2005*).

Reminiscent of the role of AChE for neuronal signal transduction in animals, it appears that, in *C. elegans*, carboxylesterases with homology to AChE have been co-opted to establish additional signal transduction pathways that are based on a modular chemical language, for inter-organismal communication, and perhaps also intra-organismal signaling. The biosynthetic functions of most of the 200 serine hydrolases in *C. elegans*, including more than 30 additional *cest* homologs, remain to be assessed, and it seems likely that this enzyme family contributes to the biosynthesis of a large number of additional, yet unidentified compounds. Similarly, the exact enzymatic roles of many families of mammalian serine hydrolases have not been investigated using HRMS-based untargeted metabolomics. Our results may motivate a systematic characterization of metazoan *cest* homologs and other serine hydrolases, with regard to their roles in metabolism and small molecule signaling, associated enzymatic mechanisms, and cellular localization.

# Materials and methods

## Key resources table

| Reagent type (species) or resource | Designation | Source or reference | Identifiers | Additional information |
|---|---|---|---|---|
| Strain, strain background *Caenorhabditis elegans* | N2 | Caenorhabditis Genetics Center (CGC) | | Wild type |
| Strain, strain background *Caenorhabditis elegans* | GH10 | David Gems | | *glo-1(zu437)* |
| Strain, strain background *Caenorhabditis elegans* | RB811 | Caenorhabditis Genetics Center (CGC) | | *glo-4(ok623)* |
| Strain, strain background *Caenorhabditis elegans* | RB2053 | Caenorhabditis Genetics Center (CGC) | | *ges-1(ok2716)* |
| Strain, strain background *Caenorhabditis elegans* | PS8031 | This work | | *cest-1.1(sy1180)* |
| Strain, strain background *Caenorhabditis elegans* | PS8032 | This work | | *cest-1.1(sy1181)* |
| Strain, strain background *Caenorhabditis elegans* | DP683 | This work | | *cest-1.1(dp683) (S213A)* |
| Strain, strain background *Caenorhabditis elegans* | PS8259 | This work | | *cest-1.1(sy1180 sy1250)* |
| Strain, strain background *Caenorhabditis elegans* | PS8260 | This work | | *cest-1.1(sy1180 sy1251)* |
| Strain, strain background *Caenorhabditis elegans* | PS8261 | This work | | *cest-1.1(sy1181 sy1252)* |
| Strain, strain background *Caenorhabditis elegans* | PS8262 | This work | | *cest-1.1(sy1181 sy1253)* |
| Strain, strain background *Caenorhabditis elegans* | PS8008 | This work | | *cest-2.2(sy1170)* |

*Continued on next page*

*Continued*

| Reagent type (species) or resource | Designation | Source or reference | Identifiers | Additional information |
|---|---|---|---|---|
| Strain, strain background *Caenorhabditis elegans* | PS8009 | This work | | *cest-2.2 (sy1171)* |
| Strain, strain background *Caenorhabditis elegans* | PS8236 | This work | | *cest-2.2(sy1170 sy1236)* |
| Strain, strain background *Caenorhabditis elegans* | PS8238 | This work | | *cest-2.2(sy1171 sy1238)* |
| Strain, strain background *Caenorhabditis elegans* | FCS02 | SunyBiotech | | *cest-2.2-mCherry* |
| Strain, strain background *Caenorhabditis elegans* | PS8116 | This work | | *cest-4(sy1192)* |
| Strain, strain background *Caenorhabditis elegans* | PS8117 | This work | | *cest-4(sy1193)* |
| Strain, strain background *Caenorhabditis elegans* | PS8781 | This work | | *cest-4(sy1192)* |
| Strain, strain background *Caenorhabditis elegans* | PS8782 | This work | | *cest-4(sy1193)* |
| Strain, strain background *Caenorhabditis elegans* | PS8783 | This work | | *cest-4(sy1194)* |
| Strain, strain background *Caenorhabditis elegans* | PS8784 | This work | | *cest-4(sy1195)* |
| Strain, strain background *Caenorhabditis elegans* | RB1804 | Caenorhabditis Genetics Center (CGC) | | *cest-6(ok2338)* |
| Strain, strain background *Caenorhabditis elegans* | PS8029 | This work | | *cest-19(sy1178)* |
| Strain, strain background *Caenorhabditis elegans* | PS8030 | This work | | *cest-19(sy1179)* |
| Strain, strain background *Caenorhabditis elegans* | PS8033 | This work | | *cest-33(sy1182)* |
| Strain, strain background *Caenorhabditis elegans* | PS8034 | This work | | *cest-33(sy1183)* |
| Strain (*Caenorhabditis briggsae*) | PS8515 | This work | | CBR-*glo-1(sy1382)* |
| Strain (*Caenorhabditis briggsae*) | PS8516 | This work | | CBR-*glo-1(sy1383)* |
| Peptide, recombinant protein | Proteinase K | New England Biolabs | | New England Biolabs: P8107S |
| Software, algorithm | Metaboseek | Metaboseek (metaboseek.com) | | Version 0.9.6 |
| Software, algorithm | GraphPad Prism | GraphPad Prism (graphpad.com) | | Version 8.4.3 |

## General information

Unless noted otherwise, all reagents were purchased from Sigma-Aldrich. All newly identified compounds were assigned four letter 'SMID's (a search-compatible, Small Molecule IDentifier) for example 'icas#3' or 'ascr#10'. For a list of all compounds referred to in the text and figures, see *Appendix 1—table 9*. The SMID database (www.smid-db.org) is an electronic resource maintained in collaboration with WormBase (www.wormbase.org). A complete list of SMIDs can be found at www.smid-db.org/browse, and example structures for different SMIDs at www.smid-db.org/smidclasses.

## BLAST analysis of *uar-1*

Amino acid sequence of *Ppa*-UAR-1 was used as previously published (*Falcke et al., 2018*). BLASTp was run from the WormBase engine at (https://wormbase.org/tools/blast_blat). E-value threshold was set to 1E0. Database was set to WS269 and species was set to *C. elegans*. Results of BLASTp search are listed in *Appendix 1—table 2*.

## Amino acid sequence alignment

hAChE was aligned with *Ppa*-UAR-1, CEST-1.1, CEST-2.2, and CEST-4 was done using T-Coffee Multiple Sequence alignment (*Notredame et al., 2000*). Protein sequences for *C. elegans* CEST proteins are from WormBase. The AChE sequence was obtained from NCBI (accession number P22303). Amino acids were colored based on chemical properties: AVFPMILW = red (small + hydrophobic), DE = blue (acidic), RHK = magenta (basic), STYHCNGQ = green (hydroxyl + sulfhydryl + amine + glycine). See *Figure 4—figure supplement 1* for results.

## Phylogenetic tree

The protein sequence of Ppa-UAR1 was submitted to an NCBI BLASTp search (*Altschul et al., 2005*) (restricted to species *C. elegans*, conditional compositional BLOSUM62, gap open cost:11, gap extension cost: 1, word size: 6) using Geneious software (Biomatters Inc). The top BLAST hits by E-value up to and including *ace-3* were selected, and only the best scoring transcript variant was kept for each protein sequence hit. A total of 28 sequences were then imported into MEGA7 (*Kumar et al., 2016*) and aligned using MUSCLE (*Edgar, 2004*) (settings: gap open penalty: −2.9, gap extend 0, hydrobicity multiplier 1.2, max. iterations 8, clustering method for all iterations: UPGMB, minimal diagonal length: 24). From this alignment, an Maximum Likelihood tree was built based on the JTT matrix-based model (*Jones et al., 1992*). Initial trees were built by applying Neighbor-Join and BioNJ algorithms to a matrix of pairwise distances estimated using a JTT model assuming uniform substitution rates across positions. Phylogeny confidence was tested using 200 bootstrap replications. The tree with the highest log likelihood (−22299.9282) is shown. At each branch, the percentage of bootstrap replicates containing the same branching event is denoted. The tree is drawn to scale, with branch lengths measured in the number of substitutions per site. The evolutionary history was inferred by using the Maximum Likelihood method based on the JTT matrix-based model (*Jones et al., 1992*). The tree with the highest log likelihood (−22299.9282) is shown. The percentage of trees in which the associated taxa clustered together is shown next to the branches. Initial tree(s) for the heuristic search were obtained automatically by applying Neighbor-Join and BioNJ algorithms to a matrix of pairwise distances estimated using a JTT model, and then selecting the topology with superior log likelihood value. The tree is drawn to scale, with branch lengths measured in the number of substitutions per site. The analysis involved 28 amino acid sequences. All positions containing gaps and missing data were eliminated. There were a total of 427 positions in the final dataset. Evolutionary analyses were conducted in MEGA7 (*Kumar et al., 2016*; *Felsenstein, 1985*).

## Nematode strains

Wild-type (N2) and *glo-1(zu437)* null animals were provided by the Caenorhabditis Genetics Center (CGC), which is funded by NIH Office of Research Infrastructure Programs (P40 OD010440). *cest-2.2* mutant strains integrating N-terminal (mCherry-*cest-2.2*) or C-terminal mCherry (*cest-2.2*-mCherry) were generated by SunyBiotech. Generation of *C. elegans and C. briggsae* null mutants and revertants as well as generation of the *cest-1.1* point mutant is described below. See *Appendix 1—table 3* for a complete list of strains used in this study.

## *C. elegans* CRISPR mutagenesis for generation of *cest* null mutants

CRISPR/Cas9 mutagenesis was performed as in *Wang et al., 2018*. Briefly, *C. elegans* strain N2 was gene-edited by insertion of a 43-base-pair insertion that disrupts translation *Appendix 1—table 8*. Independent homozygous mutants were picked among the progeny of heterozygous F1 progeny of injected hermaphrodites and given distinct unique allele names. Reversion of mutants was accomplished in the same way.

## *C. briggsae* CRISPR mutagenesis for generation of *glo-1* null mutants

The *C. briggsae glo-1* mutants *sy1382* and *sy1383* were both created using the briggsae adaptation of the STOP-IN cassette method as described in *Cohen and Sternberg, 2019* and *Wang et al., 2018*. Both strains were made using a successful insertion of the STOP-IN cassette into the middle of the first exon using the guide AACAAATCTCCGGATGATTG. To detect the insertion, we used forward primer GGGTGACCGCCCATTTATTG and reverse primer AAAGGCGCACATCTTGCTTC.

## *C. elegans* CRISPR mutagenesis for generation of the *cest-1.1(dp683)* allele encoding the S213A catalytic mutant

*cest-1.1(dp683)* was generated as previously described (**Paix et al., 2015**). Briefly, *daf-2(e1368)* mutant animals were injected with in-vitro-assembled Cas9-crRNA-tracrRNA complexes targeting *cest-1.1* and the *dpy-10* co-CRISPR gene and two 100 bp repair oligonucleotides containing the desired *cest-1.1* mutation and the *dpy-10(cn64)* co-CRISPR mutation (**Arribere et al., 2014**). Sequences of the *cest-1.1* crRNA and repair oligonucleotide are 5' acctacCGCTACTATCATAC 3' and 5' GAAA TTGAAAACTTTGGAGGAAATAAAAACAGAATTACATTGGCAGGGCATGCCGCTGGAGCAAGTA TGATAGTAGCGgtaggtcacataaatgatacatttttg 3', respectively. F1 Rol progeny of injected animals were picked and screened for the presence of the *cest-1.1(dp683)* mutation after egglay. F2 broods of F1 Rol animals that were heterozygous for *cest-1.1(dp683)* were screened for animals that were homozygous for *cest-1.1(dp683)* and either wild-type or heterozygous for *cn64* at the *dpy-10* locus. Subsequent broods were screened for wild-type *dpy-10* animals to remove the co-CRISPR mutation.

## Nematode imaging

To image, gravid adult *C. elegans* were transferred to an agarose pad on a glass slide with 10 µM of levamisole to immobilize the worms. Microscopic analysis was performed using a Leica TCS SP5 Laser Scanning Confocal Microscope. Green autofluorescence was excited at 488 nm and the emission detector was set to 490–540 nm. mCherry was excited with 561 nm and the emission detector was set to 590–650 nm. Worms were imaged using the 100x objective.

## *C. briggsae* imaging

0.5 mL of 2 µM Lysotracker Deep Red (Thermo Fisher 1 mM stock in DMSO) was added to a 6 cm NGM plate seeded with 0.1 mL of *E. coli* OP50 and incubated in the dark for 24 hr at 20℃. L4 larvae of *C. briggsae* were added to the plate and allowed to grow in the dark for 24 hr at 20℃. To image, *C. briggsae* were transferred to an agarose pad on a glass slide with 10 µM of levamisole to immobilize the worms. Microscopic analysis was performed using a Zeiss Axio Imager Z2 florescence microscope with Apotome.

## Nematode cultures, mixed stage

Culturing began by chunking *C. elegans* or *C. briggsae* onto 10 cm NGM plates (each seeded with 800 µL of OP50 *E. coli* grown to stationary phase in Lennox Broth) and incubated at 22℃. Once the food was consumed, the cultures were incubated for an additional 24 hr. Each plate was then washed with 25 mL of S-complete medium into a 125 mL Erlenmeyer flask, and 1 mL of OP50 *E. coli* was added (*E. coli* cultures were grown to stationary phase in Terrific Broth, pelleted and resuspended at 1 g wet mass per 1 mL M9 buffer), shaking at 220 RPM and 22℃. After 70 hr, cultures were centrifuged at 5000 G for 1 min. After discarding supernatant, 24 mL $H_2O$ was added, along with 6 mL bleach, 900 µL 10 M NaOH and the mixture was shaken for 3 min to prepare eggs. Eggs were centrifuged at 5000 G, the supernatant was removed, and the egg pellet washed with 35 mL M9 buffer twice and then suspended in a final volume of 5 mL M9 buffer in a 50 mL centrifuge tube. Eggs were counted and placed on a rocker and allowed to hatch as L1 larvae for 24 hr at 22℃. 70,000 L1 larvae were seeded in 25 mL cultures of S-complete with 1 mL of OP50 and incubated at 220 RPM and 22℃ in a 125 mL Erlenmeyer flask. After 72 hr, cultures were fed an additional 1 mL of OP50 and incubation continued. After an additional 48 hr, worms were spun at 1000 G 5 min and spent medium was separated from worm body pellet. Separated medium and worm pellet were flash frozen over liquid nitrogen until further processing. At least three biological replicates were grown for all mutant strains. Mutants were grown with parallel wildtype controls, and biological replicates were started on different days.

## Metabolite extraction

Lyophilized pellet and media samples were crushed and homogenized by shaking with 2.5 mm steel balls at 1300 rpm for 3 min in 30 s pulses while chilled with liquid nitrogen (SPEX sample prep miniG 1600). Thus powdered media and pellet samples were extracted with 15 mL methanol in 50 mL centrifuge tubes, rocking overnight at 22℃. Extractions were pelleted at 5000 g for 10 min at 4℃, and supernatants were transferred to 20 mL glass scintillation vials. Samples were then dried in a

SpeedVac (Thermo Fisher Scientific) vacuum concentrator. Dried materials were resuspended in 1 mL methanol and vortexed for 1 min. Samples were pelleted at 5000 g for 5 min and 22°C, and supernatants were transferred to 2 mL HPLC vials and dried in a SpeedVac vacuum concentrator. Samples were then resuspended in 200 μL of methanol, transferred into 1.7 mL Eppendorf tubes, and centrifuged at 18,000 G for 20 min at 4°C. Clarified extracts were transferred to fresh HPLC vials and stored at −20°C until analysis.

## Preparation of *exo*-metabolome samples from staged starved and fed cultures

40,000 synchronized L1 larvae were added to 125 mL Erlenmeyer flasks containing 30 mL of S-complete medium. Worms were fed with 4 mL of concentrated OP-50 and incubated at 20°C with shaking at 160 RPM for: 12 hr (L1), 24 hr (L2), 32 hr (L3), 40 hr (L4) and 58 hr (gravid adults). For preparation of starved samples, each of the stages was starved for 24 hr after reaching their desired developmental stage in S-complete without OP-50. After incubation for the desired time, liquid cultures were centrifuged (1000 x g, 22°C, 1 min) and supernatants were collected. Supernatant was separated from intact OP-50 cells by centrifuging (3000 x g, 22°C, 5 min) and the resulting supernatants (*exo*-metabolome) were lyophilized. Lyophilized samples were homogenized with a dounce homogenizer in 10 mL methanol and extracted on a stirring plate (22°C, 12 hr). The resulting suspension was centrifuged (4000 g, 22°C, 5 min) to remove any precipitate before carefully transferring to an LC-MS sample vial. Three biological replicates were started on different days.

## Mass spectrometric analysis

High resolution LC-MS analysis was performed on a Thermo Fisher Scientific Vanquish Horizon UHPLC System coupled with a Thermo Q Exactive HF hybrid quadrupole-orbitrap high-resolution mass spectrometer equipped with a HESI ion source. 1 μL of extract was injected and separated using at water-acetonitrile gradient on a Thermo Scientific Hypersil GOLD C18 column (150 mm x 2.1 mm 1.9 um particle size 175 Å pore size, Thermo Scientific) and maintained at 40°C. Solvents were all purchased from Fisher Scientific as HPLC grade. Solvent A: 0.1% formic acid in water; solvent B: 0.1% formic acid in acetonitrile. A/B gradient started at 1% B for 5 min, then from 1% to 100% B over 20 min, 100% for 5 min, then down to 1% B for 3 min. Mass spectrometer parameters: 3.5 kV spray voltage, 380°C capillary temperature, 300°C probe heater temperature, 60 sheath flow rate, 20 auxiliary flow rate, one spare gas; S-lens RF level 50.0, resolution 240,000, *m/z* range 100–1200 m/z, AGC target 3e6. Instrument was calibrated with positive and negative ion calibration solutions (Thermo-Fisher) Pierce LTQ Velos ESI pos/neg calibration solutions.

## Feature detection and characterization

LC−MS RAW files from each sample were converted to mzXML (centroid mode) using MSConvert (ProteoWizard), followed by analysis using the XCMS (*Smith et al., 2006*) analysis feature in METABOseek (metaboseek.com). Peak detection was carried out with the centWave algorithm (*Tautenhahn et al., 2008*), values set as: 4 ppm, 320 peakwidth, 3 snthresh, 3100 prefilter, FALSE fitgauss, 1 integrate, TRUE firstBaselineCheck, 0 noise, wMean mzCenterFun, −0.005 mzdiff. XCMS feature grouping values were set as: 0.2 minfrac, 2 bw, 0.002 mzwid, 500 max, 1 minsamp, FALSE usegroup. METABOseek peak filling values set as: 5 ppm_m, 5 rtw, TRUE rtrange. Resulting tables were then processed with the METABOseek Data Explorer. Molecular features were filtered for each particular null mutant against all other mutants. Filter values were set as: 10 to max minFoldOverCtrl, 15000 to max meanInt, 120 to 1500 rt, 0.95 to max Peak Quality as calculated by METABOseek. Features were then manually curated by removing isotopic and adducted redundancies. Remaining masses were put on the inclusion list for MS/MS (ddMS2) characterization. Positive and negative mode data were processed separately. In both cases we checked if a feature had a corresponding peak in the opposite ionization mode, since fragmentation spectra in different modes often provide complementary structural information. To acquire MS2 spectra, we ran a top-10 data dependent MS2 method on a Thermo QExactive-HF mass spectrometer with MS1 resolution 60,000, AGC target $1 \times 10^6$, maximum IT (injection time) 50 ms, MS2 resolution 45,000, AGC target $5 \times 10^5$, maximum IT 80 ms, isolation window 1.0 m/z, stepped NCE (normalized collision energy) 25, 50, dynamic exclusion 3 s.

## Statistical analysis

Peak integration data from HPLC-MS analysis were log-transformed (*Karpievitch et al., 2012*) prior to statistical analysis. Significance of differences between average peak areas were then assessed using unpaired t-tests.

## MS²-based molecular networking

For the differential featuresidentified above, MS$^2$ data was acquired. To generate the MS$^2$ molecular network, Metaboseek version 0.9.6 was used. Using the MS2scans function, differential features were matched with their respective MS$^2$ scan, using an *m/z* window of 5 ppm, and a retention time window of 15 s. To construct the molecular network, tolerance of the fragment peaks was set to *m/z* of 0.002 or 5 ppm, minimum number of peaks was set to 5, with a 2% noise level. Once the network was constructed, a cosine value of 0.8 was used, and the number of possible connections was constrained to 5.

## Serine hydrolase dendrogram

The serine hydrolase list was reported previously (*Chen et al., 2019*). From this list, sequences were inputted into Geneious Prime (version 2020.1.2 Biomatters). Sequences were aligned using Clustal Omega, neighbor joining alignment. Dendrogram tree was generated using the Geneious Tree Builder; Genetic distance model Jukes-Cantor, Tree build method UPGMA, no outgroup, Bootstrap resampling, random seed 508,949, 300 interactions, support threshold of 1. CEST enzymes were colored red and PPA-UAR-1 was colored blue (*Figure 1—figure supplement 1*).

## Synthetic procedures

Synthesis of iglu#1 (15). iglu#1 was synthesized as described previously (*Messaoudi et al., 2004*).

Synthesis of angl#1 (17). angl#1 was synthesized as described previously (*Coburn et al., 2013*).

**Scheme 1.** Synthesis of 2-((*tert*-butoxycarbonyl)amino)benzoic acid (Boc-AA, SI-1).

To a solution of anthranilic acid (**33**, 300 mg, 2.18 mmol) in 4 mL of THF and H$_2$O (1:1), Boc-anhydride (521 mg, 2.39 mmol) was added, and 2 M NaOH was added to the mixture until pH 10 was reached. The reaction mixture was stirred at room temperature. After 23 hr, the solution was concentrated in vacuo, and 15% citric acid aqueous solution was added until pH 4 was reached. The white precipitate was filtered off and dried under vacuum to provide 2-((*tert*-butoxycarbonyl)amino) benzoic acid (**SI-1**, 497 mg, 96%) as a white solid. $^1$H NMR, 600 MHz, chloroform-*d*: δ (ppm) 10.06 (s, 1H), 8.47 (dd, *J* = 8.7, 0.9 Hz, 1H), 8.08 (dd, *J* = 7.9, 1.5 Hz, 1H), 7.57 (dt, *J* = 7.9, 1.5 Hz, 1H), 7.03 (dt, *J* = 7.2, 1.2 Hz, 1H), 1.55 (s, 9H).

**Scheme 2.** Synthesis of *N-β*-(6-(2'-aminobenzoyl)-glucopyranosyl) indole (iglu#3, 34).

To a stirred solution of *N*-(*tert*-butoxycarbonyl)anthranilic acid (*Krueger et al., 2008*) (SI-1, 10 mg, 0.042 mmol) in dimethylformamide, 1-(3-dimethylaminopropyl)−3-ethylcarbodiimide

hydrochloride (EDC·HCl, 20.1 mg, 0.105 mmol) was added. The mixture was stirred at room temperature for 5 min, and 4-dimethylaminopyridine (DMAP, 18.1 mg, 0.105 mmol) and $N$-$\beta$-glucopyranosyl indole (iglu#1, **15**, 9.8 mg, 0.0351 mmol) were added. The reaction mixture was stirred at room temperature. After 5 hr, the mixture was concentrated in vacuo to yield a viscous oil, which was dissolved in 1.4 mL of a 5:2 mixture of dichloromethane and methanol. Trifluoroacetic acid (TFA, 0.5 mL) was added slowly and the reaction mixture was stirred at room temperature. After 3 hr, the mixture was concentrated in vacuo. Preparative HPLC provided a pure sample of iglu#3 (**34**, 0.8 mg, 5.7%). See *Appendix 1—table 4* for NMR spectroscopic data of iglu#3.

HRMS (ESI) $m/z$: $[M - H]^-$ calcd for $C_{21}H_{21}N_2O_6^-$ 397.13938; found 397.14017.

**Scheme 3.** Synthesis of $N$-$\beta$-(6-nicotinoylglucopyranosyl) indole (iglu#5, SI-2).

To a stirred solution of nicotinic acid (7.3 mg, 0.059 mmol) in a mixture of dimethylformamide and dichloromethane (1:1), EDC·HCl (28.4 mg, 0.148 mmol) was added. The mixture was stirred at room temperature for 30 min, before DMAP (18.1 mg, 0.148 mmol) and $N$-$\beta$-glucopyranosyl indole (iglu#1, **15**, 13.8 mg, 0.0494 mmol) were added. The reaction mixture was stirred at room temperature for 20 hr, the mixture was concentrated in vacuo, and flash column chromatography on silica using a gradient of 0–25% methanol in dichloromethane afforded **iglu#5** (**SI-2**, 2.5 mg, 13.9%) as a colorless oil. See *Appendix 1—table 5* for NMR spectroscopic data of iglu#5.

HRMS (ESI) $m/z$: $[M + H]^+$ calcd for $C_{20}H_{21}N_2O_6^+$ 385.13941; found 385.14038.

**Scheme 4.** Synthesis of $N$-$\beta$-(6-(2'-methylbut-2'$E$-enoyl)-glucopyranosyl) indole (iglu#7, SI-3).

To a stirred solution of tiglic acid (5.0 mg, 0.050 mmol) in a 1:1 mixture of dimethylformamide and dichloromethane, EDC·HCl (23.9 mg, 0.125 mmol) was added. The mixture was stirred at room temperature for 30 min, and DMAP (15.2 mg, 0.125 mmol) and $N$-$\beta$-glucopyranosyl indole (iglu#1, **15**, 11.6 mg, 0.0416 mmol) were added. The reaction mixture was stirred at room temperature for 22 hr and then concentrated in vacuo. Flash column chromatography on silica using a gradient of 0–30% methanol in dichloromethane afforded **iglu#7** (**SI-3**, 2.5 mg, 11.3%) as a colorless oil. See *Appendix 1—table 6* for NMR spectroscopic data of iglu#7.

HRMS (ESI) $m/z$: $[M + H]^+$ calcd for $C_{19}H_{24}NO_6^+$ 362.15981; found 362.16025.

**Scheme 5.** Synthesis of *N-β*-(6-(pyrrole-2'-carbonyl)-glucopyranosyl) indole (iglu#9, SI-4).

To a suspension of pyrrole-2-carboxylic acid (6.0 mg, 0.054 mmol) in dichloromethane, oxalyl chloride (14 µL, 0.163 mmol) was added slowly, followed by dimethylformamide (1 µL, 0.0129 mmol). The mixture was stirred at room temperature for 18 hr and then concentrated to dryness in vacuo. The residue was re-dissolved in dimethylformamide (2 mL) containing *N-β*-glucopyranosyl indole (iglu#1, **15**, 10.8 mg, 0.0387 mmol). Triethylamine (45 µL, 0.324 mmol) was added, and the reaction was stirred at 35°C for 7 days. Subsequently the mixture was concentrated in vacuo, and flash column chromatography on silica using a gradient of 0–30% methanol in dimethylformamide afforded **iglu#9** (**SI-4**, 1.5 mg, 10.4%) as a colorless oil. See *Appendix 1—table 7* for NMR spectroscopic data of iglu#9.

HRMS (ESI) *m/z*: $[M + H]^+$ calcd for $C_{19}H_{21}N_2O_6^+$ 373.13941; found 373.14026.

**Scheme 6.** Synthesis of an HPLC standard of ((2R,3S,4S,5R,6S)−6-((2-aminobenzoyl)oxy)−3,4,5-trihydroxytetrahydro-2*H*-pyran-2-yl)methyl 2-aminobenzoate (angl#3, SI-5).

To a stirred solution of Boc-AA (2 mg, 0.00 84 mmol) in dimethylformamide, 1-(3-dimethylaminopropyl)−3-ethylcarbodiimide hydrochloride (3.9 mg, 0.0203 mmol) was added. The mixture was stirred at room temperature for 5 min, and 4-dimethylaminopyridine (2.5 mg, 0.0203 mmol) and angl#1 (**17**, 2 mg, 0.0068 mmol) were added. The reaction mixture was stirred at room temperature. After 5 hr, the mixture was concentrated in vacuo. The crude product was dissolved in 0.55 mL dichloromethane and methanol (10:1), and trifluoroacetic acid (500 µL) was added slowly. The reaction mixture was stirred at room temperature for 3 hr and then was concentrated in vacuo, affording **angl#3** (**SI-5**).

HRMS (ESI) *m/z*: $[M + H]^+$ calcd for $C_{20}H_{23}N_2O_7^+$ 403.14998; found 403.15100.

**Scheme 7.** Synthesis of an HPLC standard of *N*-(*p*-aminobenzoyl)glutamate (PABA-glutamate) (**29**).

*p*-Aminobenzoic acid (Chem-Impex) (**8**) was dissolved in warm dichloromethane (DCM) containing triethylamine (0.1 eq). EDC·HCl (Amresco Biochemicals) (1 eq), and di-*tert*-butyl glutamate (1 eq) was added to the reaction mixture. *N,N*-Dimethylaminopyridine (1.1 eq) was then added and the resulting mixture was stirred at room temperature for 24 hr and then extracted with ethyl acetate. The combined extractswere dried with sodium sulfate and evaporated to dryness *in vacuo*. The crude product was dissolved in DCM, and trifluoroacetic acid was added (100 eq). The reaction was then stirred for 6 hr at room temperature. TFA and DCM were evaporated off to yield crude PABA-glutamate (**29**). [1]H NMR, 600 MHz, methanol-$d_4$: δ (ppm) 7.93 (d, *J* = 8.6 Hz, 2H), 7.37 (d, *J* = 8.5 Hz, 2H), 4.61 (dd, *J* = 5.0, 9.3 Hz, 1H), 2.09–2.28 (m, 4H).

NMR spectra appendix. NMR spectra of synthetic intermediates and newly identified metabolites.

## Acknowledgements

This research was funded by an NIH Chemical Biology Interface (CBI) Training Grant 5T32GM008500 (to B.C.), National Institutes of Health grants R35 GM131877 (to F.C.S.), and R24OD023041 (to P.W. S.). F.C.S. is a Faculty Scholar of the Howard Hughes Medical Institute. We thank WormBase for sequences, Tsui-Fen Chou for Cas9 protein, Ying (Kitty) Zhang for assistance with NMR spectroscopy, and Navid Movahed for assistance with mass spectrometry.

## Additional information

### Funding

| Funder | Grant reference number | Author |
| --- | --- | --- |
| National Institutes of Health | R35 GM131877 | Frank C Schroeder |
| National Institutes of Health | R24 OD023041 | Paul W Sternberg |
| National Institutes of Health | 5T32GM008500 | Brian J Curtis |

The funders had no role in study design, data collection and interpretation, or the decision to submit the work for publication.

### Author contributions

Henry H Le, Chester JJ Wrobel, Conceptualization, Data curation, Formal analysis, Investigation, Writing - original draft, Writing - review and editing; Sarah M Cohen, Conceptualization, Resources, Methodology; Jingfang Yu, Resources, Formal analysis; Heenam Park, Resources, Methodology; Maximilian J Helf, Software, Methodology; Brian J Curtis, Resources, Investigation; Joseph C Kruempel, Patrick J Hu, Resources; Pedro Reis Rodrigues, Data curation, Investigation; Paul W Sternberg, Conceptualization, Funding acquisition, Writing - original draft, Project administration, Writing - review and editing; Frank C Schroeder, Conceptualization, Formal analysis, Supervision, Funding acquisition, Writing - original draft, Project administration, Writing - review and editing

## Author ORCIDs

Henry H Le (iD) http://orcid.org/0000-0003-2942-2357
Jingfang Yu (iD) http://orcid.org/0000-0003-1770-5368
Paul W Sternberg (iD) https://orcid.org/0000-0002-7699-0173
Frank C Schroeder (iD) https://orcid.org/0000-0002-4420-0237

## Decision letter and Author response

Decision letter https://doi.org/10.7554/eLife.61886.sa1
Author response https://doi.org/10.7554/eLife.61886.sa2

## Additional files

### Supplementary files

• Supplementary file 1. NMR spectra appendix. NMR spectra of synthetic intermediates and newly identified metabolites.

• Transparent reporting form

### Data availability

All data generated or analysed during this study are included in the manuscript and supporting files. MS/MS data is available via MassIVE under accession number: MSV000086293.

The following dataset was generated:

| Author(s) | Year | Dataset title | Dataset URL | Database and Identifier |
|---|---|---|---|---|
| Le HH, Wrobel CJJ, Cohen SM, Yu J, Park H, Helf MJ, Curtis BJ, Kruempel JC, Rodrigues PR, Hu PJ, Sternberg PW, Schroeder FC | 2020 | Modular metabolite assembly in C. elegans depends on carboxylesterases and formation of lysosome-related organelles | https://massive.ucsd.edu/ProteoSAFe/dataset.jsp?task=715e60ce44ae4ecea2-b84e28dd336c01 | MassIVE, MSV0000 86293 |

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

## Appendix 1

## Supporting tables

**Appendix 1—table 1.** MS$^2$ data of *glo-1*-dependent features presented in this manuscript.

| | | | | | | Representative MS/MS spectra of modular glucosides. | | | |
|---|---|---|---|---|---|---|---|---|---|
| Formula | RT [min.] | Compound number | SMID | m/z (M+H) | m/z (M-H) | ms/ms fragments, positive ionization mode | ms/ms fragments, negative ionization mode | Substituents on glucose | Stable isotope labeling |
| C26H26N3O12P | 9.30 | | angl#10 | 604.13381 | 602.11813 | 105.03366 (C7 H5 O+) 120.04469 (C7 H6 O N+) | 96.96870 (H2 O4 P-) 121.02911 (C7 H5 O2-) 136.03983 (C7 H6 O2 N-) | anthranilic acid, nicotinic acid | |
| C20H22N2O7 | 8.67 | SI-5 | angl#3 | 403.14998 | 401.13542 | 120.04459 (C7 H6 O N+) 138.05496 (C7 H8 O2 N+) | | anthranilic acid, anthranilic acid | |
| C20H23N2O11P | 9.26 | 25 | angl#4 | 499.11235 | 497.09667 | 120.04463 (C7H6ON+) | 96.96868 (H2 O4 P-) 78.95800 (O3 P-) 136.03999 (C7 H6 O2 N-) 223.00078 (C6 H8 O7 P-) | anthranilic acid, anthranilic acid | |
| C19H21N2O9P | 9.59 | 22 | iglu#10 | 453.10574 | 451.09119 | 94.02916 (C5 H4 O N+) 118.06535 (C8 H8 N+) C14 H12 O2 N (C14 H12 O2 N+) | 78.95802 (O3 P-) 96.96867 (H2 O4 P-) 110.02444 (C5 H4 O2 N-) 116.05042 (C8 H6 N-) | indole, nicotinic acid | |
| C21H22NO9P | 10.79 | 23 | iglu#12 | 464.11049 | 462.09594 | 105.03382 (C7 H5 O+) 118.06538 (C8 H8 N+) 226.08620 (C14 H12 O2 N+) 348.12271 (C21 H18 O4 N+) | 78.95801 (O3 P-) 96.96865 (H2 O4 P-) | indole, benzoic acid | |
| C14H18NO8P | 6.05 | 16 | iglu#2 | 360.08541 | 358.06973 | 98.98453 (H4 O4 P+) 118.06536 (C8 H8 N+) 244.09660 (C14 H14 O3 N+) | 78.95802 (O3 P-) 96.96869 (H2 O4 P-) | indole | |
| C21H22N2O6 | 10.69 | 34 | iglu#3 | 399.15506 | 397.13938 | | 116.05032 (C8 H6 N-) 136.04002 (C7 H6 O2 N-) 215.09431 (C13 H13 O2 N-) | indole, anthranilic acid | |
| C21H23N2O9P | 10.29 | 19 | iglu#4 | 479.12252 | 477.10684 | 118.06536 (C8 H8 N+) 120.04456 (C7 H6 O N+) 138.05490 (C7 H8 O2 N+) 226.08612 (C14 H12 O2 N+) | 78.95801 (O3 P-) 96.96867 (H2 O4 P-) 116.05042 (C8 H6 N-) 136.03970 (C7 H6 O2 N-) 358.06805 (C14 H17 O8 N P-) | indole, anthranilic acid | |
| C27H26N3O10P | 10.49 | 41 | iglu#41 | 584.14398 | 582.1283 | 96.04494 (C5 H6 O N+) 120.04456 (C7 H6 O N+) 124.03937 (C6 H6 O2 N+) 166.04985 (C8 H8 O3 N+) 228.06477 (C13 H10 O3 N+) 330.03705 (C12 H13 O8 N P+) | 78.95801 (O3 P-) 96.96867 (H2 O4 P-) 122.02431 (C6 H4 O2 N-) 136.04013 (C7 H6 O2 N-) | indole, anthranilic acid, nicotinic acid | |

*Continued on next page*

*Appendix 1—table 1 continued*

**Representative MS/MS spectra of modular glucosides.**

| Formula | RT [min.] | Compound number | SMID | m/z (M+H) | m/z (M-H) | ms/ms fragments, positive ionization mode | ms/ms fragments, negative ionization mode | Substituents on glucose | Stable isotope labeling |
|---|---|---|---|---|---|---|---|---|---|
| C26H29N2O10P | 10.48 | 20 | iglu#42 | 561.16439 | 559.14871 | 83.04974 (C5 H7 O +) 118.06553 (C8 H8 N+) 120.04465 (C7 H6 O N+) 202.08635 (C12 H12 O2 N+) | 78.95805 (O3 P-) 96.96868 (H2 O4 P-)136.03995 (C7 H6 O2 N-) | indole, antranilic acid, tiglic acid | |
| C20H20N2O6 | 8.93 | SI-2 | iglu#5 | 385.13941 | 383.12373 | 106.02911 (C6 H4 O N+) 118.06535 (C8 H8 N+) 124.03936 (C6 H6 O2 N+) 268.08124 (C12 H14 O6 N+) | | indole, nicotinic acid | |
| C20H21N2O9P | 8.29 | 20 | iglu#6 | 465.10687 | 463.09119 | 106.02907 (C6 H4 O N+) 118.06532 (C8 H8 N+) 124.03942 (C6 H6 O2 N+) 226.08630 (C14 H12 O2 N+) 250.07079 (C12 H12 O5 N+) | 78.95802 (O3 P-) 96.96868 (H2 O4 P-) 122.02421 (C6 H4 O2 N-) 340.05878 (C14 H15 O7 N P-) | indole, nicotinic acid | |
| C19H23NO6 | 11.24 | SI-3 | iglu#7 | 362.15981 | 360.14413 | 83.04967 (C5 H7 O+) 101.06001 (C5 H9 O2+) 118.06536 (C8 H8 N+) 198.09097 (C13 H12 O N+) 226.08626 (C14 H12 O2 N+) | | indole, tiglic acid | |
| C19H24NO9P | 10.48 | 21 | iglu#8 | 442.12727 | 440.11159 | 83.04967 (C5 H7 O +) 101.06020 (C5 H9 O2+) 118.06538 (C8 H8 N+) 226.08621 (C14 H12 O2 N+) | 78.95798 (O3 P-) 96.96864 (H2 O4 P-) 116.05011 (C8 H6 N-) | indole, tiglic acid | |
| C19H20N2O6 | 6.33 | SI-4 | iglu#9 | 373.13941 | 371.12486 | | 110.02437 (C5 H4 O2 N-) 116.05027 (C8 H6 N-) | indole, nicotinic acid | |
| C21H27N2O11P | 4.31 | | oglu#4 | 515.14365 | 513.12797 | 120.04459 (C7 H6 O N+) 136.07550 (C8 H10 O N+) 138.05511 (C7 H8 O2 N+) 216.06795 (C12 H10 O3 N+) | 78.95781 (O3 P-) 96.96854 (H2 O4 P-) 136.03995 (C7 H6 O2 N-) 223.00067 (C6 H8 O7 P-) 376.07953 (C14 H19 O9 N P-) | octopamine, anthranilic acid | d1 from d2-L-Tyrosine |
| C18H24N2O7 | 4.79 | | sgnl#1 | 381.16563 | 379.14995 | | 217.09767 (C12 H13 O2 N2-) | n-acetylserotonin | |
| C25H29N3O8 | 7.34 | | sgnl#3 | 500.20274 | 498.18706 | 120.04427 (C7H6NO+) 160.07555 (C10H10NO+) | | n-acetylserotonin, anthranilic acid | |
| C25H30N3O11P | 7.89 | | sgnl#4 | 580.1702 | 578.15452 | 120.04459 (C7 H6 O N+) 138.05498 (C7 H8 O2 N+) 160.07590 (C10 H10 O N+) 219.11266 (C12 H15 O2 N2+) | (O3 P-) 96.96865 (H2 O4 P-) 136.04048 (C7 H6 O2 N-) 223.00072 (C6 H8 O7 P-) | n-acetylserotonin, anthranilic acid | |

*Continued on next page*

*Appendix 1—table 1 continued*

| | | | | | | **ms/ms fragments, positive ionization mode** | **ms/ms fragments, negative ionization mode** | **Substituents on glucose** | **Stable isotope labeling** |
|---|---|---|---|---|---|---|---|---|---|
| **Formula** | **RT [min.]** | **Compound number** | **SMID** | **m/z (M +H)** | **m/z (M-H)** | | | | |
| C29H33N2O11P | 8.22 | | tyglu#12 | 617.1906 | 615.17492 | 120.04458 (C7 H6 O N+) 238.08728 (C15 H12 O2 N+) | 78.95803 (O3 P-) 96.96867 (H2 O4 P-) 136.04008 (C7 H6 O2 N-) 135.04503 (C8 H7 O2-) 360.08469 (C14 H19 O8 N P-) 478.12738 (C29 H20 O6 N-) | tyramine, anthranilic acid, phenylacetic acid | d2 from d2-L-Tyrosine |
| C26H35N2O11P | 7.92 | | tyglu#14 | 583.20625 | 581.19057 | 109.02870 (C6 H5 O2+) 120.04459 (C7 H6 O N+) 138.05489 (C7 H8 O2 N+) 204.10226 (C12 H14 O2 N+) 257.12808 (C15 H17 O2 N2+) 348.14429 (C18 H22 O6 N+) | 78.95802 (O3 P-) 96.96866 (H2 O4 P-) 101.05991 (C5 H9 O2-) 136.04047 (C7 H6 O2 N-) 444.14252 (C19 H27 O9 N P-) | tyramine, anthranilic acid, (iso)valeric acid | |
| C28H31N2O11P | 7.97 | | tyglu#16 | 603.17495 | 601.15927 | 105.03380 (C7 H5 O+) 120.04455 (C7 H6 O N+) 138.05487 (C7 H8 O2 N+) 224.07047 (C14 H10 O2 N+) 257.12775 (C15 H17 O2 N2+) 368.11160 (C20 H18 O6 N+) | 78.95805 (O3 P-) 96.96869 (H2 O4 P-) 121.02914 (C7 H5 O2-) 136.03978 (C7 H6 O2 N-) 464.11099 (C21 H23 O9 N P-) | tyramine, anthranilic acid, carboxy-benzyl | d2 from d2-L-Tyrosine |
| C21H27N2O10P | 5.40 | | tyglu#2 | 499.14874 | 497.13306 | 120.04459 (C7 H6 O N+) 138.05487 (C7 H8 O2 N+) 138.09137 (C8 H12 O N+) 257.12814 (C15 H17 O2 N2+) 264.08633 (C13 H14 O5 N+) | 78.95802 (O3 P-) 96.96870 (H2 O4 P-) 136.04005 (C7 H6 O2 N-) 223.00053 (C6 H8 O7 P-) 360.08472 (C14 H19 O8 N P-) | tyramine, anthranilic acid | d2 from d2-L-Tyrosine |
| C28H32N3O11P | 7.65 | 26 | tyglu#4 | 618.18585 | 616.17017 | 120.04459 (C7 H6 O N+) 138.09137 (C8 H12 O N+) | 78.95802 (O3 P-) 96.96867 (H2 O4 P-) 136.03989 (C7 H6 O2 N-) 479.12198 (C21 H24 O9 N2 P-) | tyramine, anthranilic acid (x2) | |
| C27H30N3O11P | 6.55 | | tyglu#6 | 604.1702 | 602.15452 | 106.02901 (C6 H4 O N+) 120.04460 (C7 H6 O N+) 124.03939 (C6 H6 O2 N+) 138.05513 (C7 H8 O2 N+) 166.04988 (C8 H8 O3 N+) 257.12781 (C15 H17 O2 N2+) | 78.95781 (O3 P-) 96.96851 (H2 O4 P-) 223.00017 (C6 H8 O7 P-) 381.09375 (C16 H17 O9 N2-) 534.17279 (C22 H33 O12 N P-) | tyramine, anthranilic acid, nicotinic acid | d2 from d2-L-Tyrosine |
| C26H33N2O11P | 7.67 | | tyglu#8 | 581.1906 | 579.17492 | 83.04968 (C5 H7 O+) 120.04460 (C7 H6 O N+) 138.05479 (C7 H8 O2 N+) 257.12848 (C15 H17 O2 N2+) | 78.95779 (O3 P-) 96.96852 (H2 O4 P-) 99.04408 (C5 H7 O2-) 136.03972 (C7 H6 O2 N-) 442.12637 (C19 H25 O9 N P-) | tyramine, anthranilic acid, tiglic acid | d2 from d2-L-Tyrosine |

**Appendix 1—table 2.** BLASTp results from the WormBase BLAST engine when searching against the amino acid sequence of UAR-1 and CRISPR/Cas9 targets for this study (red).

| Sequence | Score | E-value |
|---|---|---|
| C01B10.10 | 280 | 2e-75 |
| C01B10.4a | 260 | 2e-69 |
| T22D1.11 | 248 | 7e-66 |
| C42D4.2 | 233 | 4e-61 |
| C17H12.4 | 231 | 1e-60 |
| C23H4.4a | 225 | 8e-59 |
| C23H4.7 | 199 | 6e-51 |
| C23H4.3 | 194 | 1e-49 |
| E01G6.3 | 193 | 3e-49 |
| C23H4.2 | 168 | 1e-41 |
| T02B5.1 | 157 | 2e-38 |
| F15A8.6a | 154 | 1e-37 |
| F15A8.6b | 154 | 1e-37 |
| ZC376.3 | 153 | 3e-37 |
| T02B5.3 | 150 | 2e-36 |
| ZC376.2b | 148 | 1e-35 |
| ZC376.2a | 147 | 2e-35 |
| F56C11.6b | 141 | 1e-33 |
| F56C11.6a | 137 | 2e-32 |
| Y71H2AM.13 | 136 | 5e-32 |
| ZC376.1 | 135 | 1e-31 |
| R173.3 r | 129 | 6e-30 |
| T07H6.1a | 127 | 2e-29 |
| T28C12.4a | 124 | 1e-28 |
| T28C12.4b | 124 | 2e-28 |
| K07C11.4 | 119 | 6e-27 |
| R12A1.4 | 118 | 1e-26 |
| K11G9.2 | 116 | 4e-26 |
| 02B12.4 | 115 | 8e-26 |
| Y75B8A.3 | 114 | 3e-25 |
| Y48B6A.8 | 113 | 4e-25 |
| F13H6.3 | 111 | 2e-24 |
| Y48B6A.7 | 109 | 5e-24 |
| 09B12.1 | 108 | 9e-24 |
| K11G9.1 | 108 | 2e-23 |
| ZC376.2c | 105 | 7e-23 |
| F07C4.12b | 105 | 7e-23 |
| C52A10.1 | 101 | 1e-21 |
| Y44E3A.2 | 101 | 2e-21 |
| K11G9.3 | 99 | 1e-20 |
| C52A10.2 | 97 | 3e-20 |

*Continued on next page*

*Appendix 1—table 2 continued*

| Sequence | Score | E-value |
|---|---|---|
| C40C9.5d | 96 | 6e-20 |
| C40C9.5b | 96 | 6e-20 |
| C40C9.5a | 96 | 6e-20 |
| F55D10.3 | 96 | 1e-19 |
| C40C9.5f | 94 | 2e-19 |
| C01B10.4b | 94 | 2e-19 |
| C40C9.5g | 94 | 2e-19 |
| C40C9.5c | 94 | 3e-19 |
| C40C9.5e | 94 | 3e-19 |
| B0238.7 | 93 | 4e-19 |
| B0238.1 | 92 | 1e-18 |
| F55F3.2b | 83 | 6e-16 |
| F55F3.2a | 83 | 7e-16 |
| C23H4.4b | 50 | 5e-06 |
| Y43F8A.3a | 42 | 0.002 |
| Y43F8A.3b | 35 | 0.18 |

**Appendix 1—table 3.** List of *C. elegans* strains used in this study.

| Strain name | Identifier | Description | Associated metabolites |
|---|---|---|---|
| PS8031 | *cest-1.1(sy1180)* | *cest-1.1* null | uglas#1 uglas#11 |
| PS8032 | *cest-1.1(sy1181)* | *cest-1.1* null | uglas#1 uglas#11 |
| PS8259 | *cest-1.1(sy1180 sy1250)* | *cest-1.1* null reverted to WT sequence | uglas#1 uglas#11 |
| PS8260 | *cest-1.1(sy1180 sy1251)* | *cest-1.1* null reverted to WT sequence | uglas#1 uglas#11 |
| PS8261 | *cest-1.1(sy1181 sy1252)* | *cest-1.1* null reverted to WT sequence | uglas#1 uglas#11 |
| PS8262 | *cest-1.1(sy1181 sy1253)* | *cest-1.1* null reverted to WT sequence | uglas#1 uglas#11 |
| PS8008 | *cest-2.2(sy1170)* | *cest-2.2* null | ascr#8, ascr#81, ascr#82 |
| PS8009 | *cest-2.2(sy1171)* | *cest-2.2* null | ascr#8, ascr#81, ascr#82 |
| PS8236 | *cest-2.2(sy1170 sy1236)* | *cest-2.2* null reverted to WT sequence | ascr#8, ascr#81, ascr#82 |
| PS8238 | *cest-2.2(sy1171 sy1238)* | *cest-2.2* null reverted to WT sequence | ascr#8, ascr#81, ascr#82 |
| PS8116 | *cest-4(sy1192)* | *cest-4* null | iglu class modular glucosides |
| PS8117 | *cest-4(sy1193)* | *cest-4* null | iglu class modular glucosides |
| JJ1271 | *glo-1(zu437)* | *glo-1* null | Most known modular ascarosides/ glucosides |
| PS8781 | *cest-4(sy1192)* | *cest-4* null reverted to WT sequence | iglu class modular glucosides |
| PS8782 | *cest-4(sy1193)* | *cest-4* null reverted to WT sequence | iglu class modular glucosides |
| PS8783 | *cest-4(sy1194)* | *cest-4* null reverted to WT sequence | iglu class modular glucosides |
| PS8784 | *cest-4(sy1195)* | *cest-4* null reverted to WT sequence | iglu class modular glucosides |
| PS8515 | CBR-*glo-1*-A (*sy1382*) | *C. briggsae glo-1* null | Most known modular ascarosides/ glucosides |

*Continued on next page*

*Appendix 1—table 3 continued*

| Strain name | Identifier | Description | Associated metabolites |
|---|---|---|---|
| PS8516 | CBR-*glo-1*-B (*sy1383*) | C. briggsae *glo-1* null | Most known modular ascarosides/glucosides |
| PS8029 | *cest-19(sy1178)* | *cest-19* null | Undetermined |
| PS8030 | *cest-19(sy1179)* | *cest-19* null | Undetermined |
| PS8033 | *cest-33(sy1182)* | *cest-33* null | Undetermined |
| PS8034 | *cest-33(sy1183)* | *cest-33* null | Undetermined |
| RB2053 | *ges-1 (ok2716)* | *ges-1* null | Undetermined |
| RB1804 | *cest-6(ok2338)* | *cest-6* null | Undetermined |
| DP683 | *cest-1.1(dp683)* | *cest-1.1* (S213A) point mutant | uglas#1 uglas#11 |
| FCS02 | *cest-2.2*-mCherry | *cest-2.2* C-terminal mCherry | ascr#8, ascr#81, ascr#82 |

**Appendix 1—table 4.** NMR spectroscopic data for iglu#3 (34).
$^1$H (600 MHz), HSQC, and HMBC NMR spectroscopic data were acquired in methanol-$d_4$. Chemical shifts were referenced to $\delta$(C$\underline{H}$D$_2$OD)=3.31 ppm and $\delta$($^{13}$CHD$_2$OD)=49.00 ppm.

| Position | $\delta$ $^{13}$C [ppm] | $\delta$ $^1$H ([ppm] $J_{HH}$[Hz]) | HMBC |
|---|---|---|---|
| 1 | 86.9 | 5.51 ($J_{1,2}$ = 9.3) | C-2, C-3, C-5, C-2′, C-9′ |
| 2 | 73.0 | 3.99 ($J_{2,3}$ = 9.0) | C-1, C-3 |
| 3 | 78.7 | 3.65 ($J_{3,4}$ = 9.0) | C-4 |
| 4 | 71.3 | 3.64 ($J_{4,5}$ = 9.1) | C-3 |
| 5 | 77.5 | 3.91 ($J_{5,6a}$ = 5.5) | C-4 |
| 6a | 64.1 | 4.43 ($J_{6a,6b}$ = 12.1) | C-5, C-1″ |
| 6b | | 4.67 ($J_{5,6b}$ = 2.2) | C-4, C-1″ |
| 2′ | 126.3 | 7.37 ($J_{2′,3′}$=3.3) | C-1 (weak), C-3′, C-4′, C-8′ (weak), C-9′ |
| 3′ | 102.9 | 6.48 | |
| 4′ | 130.4 | | |
| 5′ | 121.4 | 7.52 ($J_{5′,6′}$=8.0) | C-3′, C-7′, C-9′ |
| 6′ | 120.8 | 7.03 ($J_{6′,7′}$=7.4, $J_{3′,6′}$=1.1) | C-4′, C-8′ |
| 7′ | 122.4 | 7.06 | C-5′, C-9′ |
| 8′ | 111.5 | 7.53 | C-4′, C-6′ |

*Continued on next page*

| | | | |
|---|---|---|---|
| 9′ | 137.5 | | |
| 1″ | 168.6 | | |
| 2″ | 112.8 | | |
| 3″ | 132.1 | 7.90 ($J_{3'',4''}$=8.2, $J_{3'',5''}$=1.4) | C-1″, C-5″, C-7″ |
| 4″ | 118.2 | 6.73 ($J_{4'',5''}$=7.6) | C-2″, C-6″ |
| 5″ | 135.0 | 7.32 ($J_{5'',6''}$=7.8) | C-3″, C-7″ |
| 6″ | 118.6 | 6.84 | C-2″, C-4″ |
| 7″ | 149.9 | | |

**Appendix 1—table 5.** NMR spectroscopic data for iglu#5 (SI-2).
[1]H (600 MHz), HSQC, and HMBC NMR spectroscopic data were acquired in methanol-$d_4$. Chemical shifts were referenced to δ(C$\underline{H}$D$_2$OD)=3.31 ppm and δ([13]$\underline{C}$HD$_2$OD)=49.00 ppm.

| Position | δ [13]C [ppm] | δ [1]H ([ppm] $J_{HH}$[Hz]) | HMBC |
|---|---|---|---|
| 1 | 86.9 | 5.51 ($J_{1,2}$ = 9.2) | C-2, C-3, C-5, C-2′, C-9′ |
| 2 | 73.0 | 4.00 ($J_{2,3}$ = 9.0) | C-1, C-3 |
| 3 | 78.7 | 3.65 ($J_{3,4}$ = 9.0) | C-4 |
| 4 | 71.4 | 3.63 ($J_{4,5}$ = 8.9) | C-3 |
| 5 | 77.4 | 3.95 ($J_{5,6a}$ = 5.8) | C-4 |
| 6a | 65.3 | 4.51 ($J_{6a,6b}$ = 12.1) | C-4, C-5, C-1″ |
| 6b | | 4.75 ($J_{5,6b}$ = 2.3) | C-4, C-5, C-1″ |
| 2′ | 126.4 | 7.37 ($J_{2',3'}$=3.5) | C-3′, C-4′, C-9′ |
| 3′ | 103.1 | 6.47 | C-2′, C-4′, C-9′ |
| 4′ | 130.5 | | |
| 5′ | 121.4 | 7.51 ($J_{5',6'}$=7.9) | C-4′, C-6′, C-9′ |
| 6′ | 120.8 | 7.01 ($J_{6',7'}$=7.5, $J_{3',6'}$=1.2) | C-4′, C-8′ |
| 7′ | 122.5 | 7.05 | C-4′, C-5′, C-8′, C-9′ |
| 8′ | 111.4 | 7.49 | C-4′, C-6′ |
| 9′ | 137.6 | | |
| 1″ | 165.8 | | |
| 2″ | 127.7 | | |
| 3″ | 150.8 | 9.12 ($J_{3'',6''}$=0.5, $J_{3'',7''}$=2.0) | C-2″, C-5″, C-7″ |

*Continued on next page*

| | | | |
|---|---|---|---|
| 5″ | 153.7 | 8.74 ($J_{5'',6'''}$=4.9, $J_{5''',7''}$=1.7) | C-3″, C-6″, C-7″ |
| 6″ | 125.1 | 7.54 ($J_{6''',7''}$=8.0) | C-2″, C-5″ |
| 7″ | 138.9 | 8.37 | C-1″, C-2″, C-5″ |

**Appendix 1—table 6.** NMR spectroscopic data for iglu#7 (SI-3).
$^1$H (600 MHz), HSQC, and HMBC NMR spectroscopic data were acquired in methanol-$d_4$. Chemical shifts were referenced to δ(C$\underline{H}$D$_2$OD)=3.31 ppm and δ($^{13}$CHD$_2$OD)=49.00 ppm.

| Position | δ $^{13}$C [ppm] | δ $^1$H ([ppm] $J_{HH}$[Hz]) | HMBC |
|---|---|---|---|
| 1 | 86.9 | 5.46 ($J_{1,2}$ = 9.1) | C-2, C-3, C-5, C-2′, C-9′ |
| 2 | 73.2 | 3.96 ($J_{2,3}$ = 9.0) | C-1, C-3 |
| 3 | 78.9 | 3.61 ($J_{3,4}$ = 9.0) | C-2, C-4 |
| 4 | 71.4 | 3.55 ($J_{4,5}$ = 9.6) | C-3, C-5, C-6 |
| 5 | 77.6 | 3.81 ($J_{5,6a}$ = 5.6) | C-1 (weak), C-3, C-4 |
| 6a | 64.5 | 4.27 ($J_{6a,6b}$ = 11.9) | C-4, C-5, C-1″ |
| 6b | | 4.49 ($J_{5,6b}$ = 2.2) | C-4, C-5, C-1″ |
| 2′ | 126.6 | 7.35 ($J_{2',3'}$=3.5) | C-1 (weak), C-3′, C-4′, C-5′ (weak), C-8′ (weak), C-9′ |
| 3′ | 103.2 | 6.48 | |
| 4′ | 130.6 | | |
| 5′ | 121.6 | 7.53 ($J_{5',6'}$=7.9) | C-3′, C-7′, C-9′ |
| 6′ | 120.9 | 7.05 ($J_{6',7'}$=7.5, $J_{3',6'}$=1.1) | C-4′, C-8′, C-9′ (weak) |
| 7′ | 122.5 | 7.11 | C-5′, C-8′ (weak), C-9′ |
| 8′ | 111.7 | 7.50 | C-4′, C-6′ |
| 9′ | 137.6 | | |
| 1″ | 169.2 | | |
| 2″ | 129.3 | | |
| 3″ | 138.9 | 6.87 ($J_{3'',4''}$=6.8) | C-1″, C-4″, C-5″ |
| 4″ | 14.2 | 1.79 | C-2″, C-3″ |
| 5″ | 11.9 | 1.81 | C-1″, C-2″, C-3″ |

**Appendix 1—table 7.** NMR spectroscopic data for iglu#9 (SI-4).
$^1$H (600 MHz), HSQC, and HMBC NMR spectroscopic data were acquired in methanol-$d_4$. Chemical shifts were referenced to δ(C$\underline{H}$D$_2$OD)=3.31 ppm and δ($^{13}$CHD$_2$OD)=49.00 ppm.

| Position | δ $^{13}$C [ppm] | δ $^1$H ([ppm] $J_{HH}$[Hz]) | HMBC |
|---|---|---|---|
| 1 | 86.9 | 5.47 ($J_{1,2}$ = 9.1) | C-2, C-3, C-5, C-2′, C-9′ |
| 2 | 73.2 | 3.96 ($J_{2,3}$ = 9.0) | C-1, C-3 |
| 3 | 78.7 | 3.62 ($J_{3,4}$ = 9.8) | C-4 |
| 4 | 71.3 | 3.61 ($J_{4,5}$ = 9.7) | C-3 |
| 5 | 77.9 | 3.86 ($J_{5,6a}$ = 5.7) | |
| 6a | 63.9 | 4.38 ($J_{6a,6b}$ = 11.9) | C-5, C-1″ |
| 6b | | 4.68 ($J_{5,6b}$ = 2.1) | C-4, C-1″ |
| 2′ | 126.6 | 7.36 ($J_{2′,3′}$=3.4) | C-3′, C-4′, C-9′ |
| 3′ | 103.1 | 6.47 | C-2′, C-4′, C-9′ |
| 4′ | 130.6 | | |
| 5′ | 121.4 | 7.52 ($J_{5′,6′}$=7.8) | C-7′, C-9′ |
| 6′ | 120.8 | 7.02 ($J_{6′,7′}$=7.3, $J_{3′,6′}$=1.2) | C-4′, C-8′ |
| 7′ | 122.4 | 7.05 | C-5′, C-9′ |
| 8′ | 111.6 | 7.50 | C-4′, C-6′ |
| 9′ | 137.4 | | |
| 1″ | 162.4 | | |
| 2″ | 123.0 | | |
| 4″ | 124.7 | 6.96 ($J_{4″,5″}$=2.5, $J_{4″,6″}$=1.4) | C-2″, C-5″, C-6″ |
| 5″ | 110.6 | 6.20 ($J_{5″,6″}$=3.8) | C-2″(weak), C-4″(weak) |
| 6″ | 116.8 | 6.90 | C-2″, C-4″, C-5″ |

**Appendix 1—table 8.** DNA oligonucleotides used for this study.

| Target gene | Sequence name | Strain | Allelle | Guide sequence | ssDNA repair oligonucleotide sequence |
|---|---|---|---|---|---|
| cest-1.1 | T02B5.1 | PS8031, PS8032 | sy1180, sy1181 | ACTCCTTCCCATGATTTCGG | TATTCATTTGTTACCAAAACTCCTTCCCATGATTTG CTAGCTTATCACTTAGTCACCTCTGCTC TGGACAAA CTTCCCCGGTGGACGGGGTTTTCGATA TCGAAGGTCTCCAATTG |
| cest-2.2 | ZC376.2 | PS8008, PS8009 | sy1170, sy1171 | GGAGGCGAAGGAGTATAAAG | CCCTGGGACGGAG TTTTGGAGGCGAAGGAGTATA GGGAAGTTTGTCCAGAGCAGAGGTGAC TAAGTGATAA GCTAGCAAGCGGCTTGTATGAGTGATCA-GAAGTAAGAGATA |

*Continued on next page*

*Appendix 1—table 8 continued*

| Target gene | Sequence name | Strain | Allelle | Guide sequence | ssDNA repair oligonucleotide sequence |
|---|---|---|---|---|---|
| *cest-4* | C17H12.4 | PS8116, PS8117 | *sy1192, sy1193* | ACTCCGGTCCA TTTCTCAGG | CATACCTTTTGCATTTCTCACTCCGGTCCA TTTCTCGCTAGC TTATCACTTAGTCACCTCTGCTCTGGA-CAAACTTCCCAGGCGG TTCTGGTTTTTGAAATCTTAATTTTCCAA TTG |

**Appendix 1—table 9.** List of all modular metabolites referred to in the text and Figures.

| Compound number | SMID ID | IUPAC Name | Evidence | Structure |
|---|---|---|---|---|
| 1 | icas#3 | (*R*)−8-(((2*R*,3*R*,5*R*,6*S*)−5-((1*H*-indole-3-carbonyl)oxy)−3-hydroxy-6-methyltetrahydro-2*H*-pyran-2-yl)oxy)nonanoic acid | Previously identified via synthesis (*Srinivasan et al., 2012*) |  |
| 2 | ascr#8 | 4-((*R*,*E*)−6-(((2*R*,3*R*,5*R*,6*S*)−3,5-dihydroxy-6-methyltetrahydro-2*H*-pyran-2-yl)oxy)hept-2-enamido)benzoic acid | Previously identified via synthesis (*Pungaliya et al., 2009*) |  |
| 3 | uglas#11 | (2*R*,3*R*,4*S*,5*R*,6*R*)−5-hydroxy-6-(hydroxymethyl)−4-(phosphonooxy)−2-(2,6,8-trioxo-1,2,6,7,8,9-hexahydro-3*H*-purin-3-yl)tetrahydro-2*H*-pyran-3-yl (*R*)−6-(((2*R*,3*R*,5*R*,6*S*)−3,5-dihydroxy-6-methyltetrahydro-2*H*-pyran-2-yl)oxy)heptanoate | Previously identified via synthesis (*Curtis et al., 2020*) |  |
| 4 | ubas#3 | (*R*)−4-(((2*R*,3*R*,5*R*,6*S*)−3-hydroxy-6-methyl-5-(((*R*)−2-methyl-3-ureidopropanoyl)oxy)tetrahydro-2*H*-pyran-2-yl)oxy)pentanoic acid | Previously inferred via tandem mass spectrometry (*Falcke et al., 2018*) |  |

*Continued on next page*

*Appendix 1—table 9 continued*

| Compound number | SMID ID | IUPAC Name | Evidence | Structure |
|---|---|---|---|---|
| 5 | ascr#1 | (R)−6-(((2R,3R,5R,6S)−3,5-dihydroxy-6-methyltetrahydro-2H-pyran-2-yl)oxy)heptanoic acid | Previously identified via NMR and synthesis (*Jeong et al., 2005*) | |
| 6 | gluric#1 | 3-((2R,3R,4S,5S,6R)−3,4,5-trihydroxy-6-(hydroxymethyl)tetrahydro-2H-pyran-2-yl)−7,9-dihydro-1H-purine-2,6,8 (3H)-trione | Previously identified via synthesis (*Curtis et al., 2020*) | |
| 7 | ascr#7 | (R,E)−6-(((2R,3R,5R,6S)−3,5-dihydroxy-6-methyltetrahydro-2H-pyran-2-yl)oxy)hept-2-enoic acid | Previously identified via synthesis (*Pungaliya et al., 2009*) | |
| 8 | PABA | 4-Aminobenzoic acid | Commercial product (Sigma-Aldrich) | |
| 9 | ascr#3 | (R,E)−8-(((2R,3R,5R,6S)−3,5-dihydroxy-6-methyltetrahydro-2H-pyran-2-yl)oxy)non-2-enoic acid | Previously identified via synthesis (*Butcher et al., 2007*) | |
| 10 | ascr#10 | (R)−8-(((2R,3R,5R,6S)−3,5-dihydroxy-6-methyltetrahydro-2H-pyran-2-yl)oxy)nonanoic acid | Previously identified via synthesis (*Srinivasan et al., 2012*) | |

*Continued on next page*

*Appendix 1—table 9 continued*

| Compound number | SMID ID | IUPAC Name | Evidence | Structure |
|---|---|---|---|---|
| 11 | | 1*H*-indole-3-carboxylic acid | Commercial product (Sigma-Aldrich) |  |
| 12 | | (*R*)—4-((2-hydroxy-2-(4-hydroxyphenyl)ethyl)amino)—4-oxobutanoic acid | Identified via synthesis (This manuscript) |  |
| 13 | iglas#1 | ((2*R*,3*S*,4*S*,5*R*,6*R*)—3,4,5-trihydroxy-6-(1*H*-indol-1-yl)tetrahydro-2*H*-pyran-2-yl)methyl (*R*)—6-(((2*R*,3*R*,5*R*,6*S*)—3,5-dihydroxy-6-methyltetrahydro-2*H*-pyran-2-yl)oxy)heptanoate | Previously identified via synthesis (*Artyukhin et al., 2018*) |  |
| 14 | glas#10 | (2*S*,3*R*,4*S*,5*S*,6*R*)—3,4,5-trihydroxy-6-(hydroxymethyl)tetrahydro-2*H*-pyran-2-yl (*R*)—8-(((2*R*,3*R*,5*R*,6*S*)—3,5-dihydroxy-6-methyltetrahydro-2*H*-pyran-2-yl)oxy)nonanoate | Previously identified via NMR and synthesis (*Coburn et al., 2013*) |  |
| 15 | iglu#1 | (2*R*,3*S*,4*S*,5*R*,6*R*)—2-(hydroxymethyl)—6-(1*H*-indol-1-yl)tetrahydro-2*H*-pyran-3,4,5-triol | Previously identified via NMR and synthesis (*Coburn et al., 2013*) |  |
| 16 | iglu#2 | (2*R*,3*R*,4*S*,5*R*,6*R*)—3,5-dihydroxy-2-(hydroxymethyl)—6-(1*H*-indol-1-yl)tetrahydro-2*H*-pyran-4-yl dihydrogen phosphate | Previously identified via NMR (*Coburn et al., 2013*) |  |

*Continued on next page*

*Appendix 1—table 9 continued*

| Compound number | SMID ID | IUPAC Name | Evidence | Structure |
|---|---|---|---|---|
| 17 | angl#1 | (2S,3R,4S,5S,6R)—3,4,5-trihydroxy-6-(hydroxymethyl) tetrahydro-2H-pyran-2-yl 2-aminobenzoate | Previously identified via NMR and synthesis (*Coburn et al., 2013*) |  |
| 18 | angl#2 | (2S,3R,4S,5R,6R)—3,5-dihydroxy-6-(hydroxymethyl)—4-(phosphonooxy) tetrahydro-2H-pyran-2-yl 2-aminobenzoate | Previously identified via NMR (*Coburn et al., 2013*) |  |
| 19 | iglu#4 | (2R,3R,4S,5R,6R)—3,5-dihydroxy-6-(1H-indol-1-yl)—4-(phosphonooxy) tetrahydro-2H-(pyran-2-yl)methyl 2-aminobenzoate | Proposed structure, based on identification of non-phosphorylated derivative (**34**) via synthesis (This manuscript) |  |
| 20 | iglu#6 | ((2R,3R,4S,5R,6R)—3,5-dihydroxy-6-(1H-indol-1-yl)—4-(phosphonooxy) tetrahydro-2H-pyran-2-yl)methyl nicotinate | Proposed structure, based on identification of non-phosphorylated derivative (**SI-2**) via synthesis (This manuscript) |  |
| 21 | iglu#8 | ((2R,3R,4S,5R,6R)—3,5-dihydroxy-6-(1H-indol-1-yl)—4-(phosphonooxy) tetrahydro-2H-pyran-2-yl)methyl (E)—2-methylbut-2-enoate | Proposed structure, based on identification of non-phosphorylated derivative (**SI-3**) via synthesis (This manuscript) |  |

*Continued on next page*

*Appendix 1—table 9 continued*

| Compound number | SMID ID | IUPAC Name | Evidence | Structure |
|---|---|---|---|---|
| 22 | iglu#10 | ((2R,3R,4S,5R,6R)−3,5-dihydroxy-6-(1H-indol-1-yl)−4-(phosphonooxy)tetrahydro-2H-pyran-2-yl)methyl 1H-pyrrole-2-carboxylate | Proposed structure, based on identification of non-phosphorylated derivative (**SI-4**) via synthesis (This manuscript) |  |
| 23 | iglu#12 | ((2R,3R,4S,5R,6R)−3,5-dihydroxy-6-(1H-indol-1-yl)−4-(phosphonooxy)tetrahydro-2H-pyran-2-yl)methyl benzoate | Proposed structure. Inferred via tandem mass spectrometry (This manuscript) |  |
| 24 | iglu#41 | (2R,3R,4S,5R,6R)−6-(((2-aminobenzoyl)oxy)methyl)−5-hydroxy-2-(1H-indol-1-yl)−4-(phosphonooxy)tetrahydro-2H-pyran-3-yl 1H-pyrrole-2-carboxylate | Proposed structure. Inferred from iglu#3 (**34**) via tandem mass spectrometry (This manuscript) |  |
| 25 | angl#4 | ((2R,3R,4S,5R,6S)−6-((2-aminobenzoyl)oxy)−3,5-dihydroxy-4-(phosphonooxy)tetrahydro-2H-pyran-2-yl)methyl 2-aminobenzoate | Proposed structure. Inferred from angl#3 (**SI 5**) via tandem mass spectrometry (This manuscript) |  |
| 26 | tyglu#4 | ((2R,3R,4S,5R,6R)−5-((2-aminobenzoyl)oxy)−3-hydroxy-6-((4-(2-aminoethyl)phenoxy)−4-(phosphonooxy)tetrahydro-2H-pyran-2-yl))methyl 2-aminobenzoate | Proposed structure. Initially described (*O'Donnell et al., 2020*) and further inferred via tandem mass spectrometry (This manuscript) |  |
| 27 | ascr#81 | (4-((R,E)−6-(((2R,3R,5R,6S)−3,5-dihydroxy-6-methyltetrahydro-2H-pyran-2-yl)oxy)hept-2-enamido)benzoyl)-L-glutamic acid | Identified via synthesis (*Artyukhin et al., 2018*) |  |

*Continued on next page*

Appendix 1—table 9 continued

| Compound number | SMID ID | IUPAC Name | Evidence | Structure |
|---|---|---|---|---|
| 28 | ascr#82 | ((S)−4-carboxy-4-(4-((R,E)−6-(((2R,3R,5R,6S)−3,5-dihydroxy-6-methyltetrahydro-2H-pyran-2-yl)oxy)hept-2-enamido)benzamido)butanoyl)-L-glutamic acid | Previously inferred via tandem mass spectrometry (*Artyukhin et al., 2018*) |  |
| 29 | PABA-glu | (4-aminobenzoyl)-L-glutamic acid | Identified via synthesis (This manuscript) |  |
| 30 | uglas#1 | (2R,3R,4S,5S,6R)−4,5-dihydroxy-6-(hydroxymethyl)−2-(2,6,8-trioxo-1,2,6,7,8,9-hexahydro-3H-purin-3-yl)tetrahydro-2H-pyran-3-yl (R)−6-(((2R,3R,5R,6S)−3,5-dihydroxy-6-methyltetrahydro-2H-pyran-2-yl)oxy)heptanoate | Identified via synthesis (*Curtis et al., 2020*) |  |
| 31 | uglas#14 | ((2R,3S,4S,5R,6R)−3,4,5-trihydroxy-6-(2,6,8-trioxo-1,2,6,7,8,9-hexahydro-3H-purin-3-yl)tetrahydro-2H-pyran-2-yl)methyl (R)−6-(((2R,3R,5R,6S)−3,5-dihydroxy-6-methyltetrahydro-2H-pyran-2-yl)oxy)heptanoate | Identified via synthesis (*Curtis et al., 2020*) |  |
| 32 | uglas#15 | ((2R,3R,4S,5R,6R)−3,5-dihydroxy-4-(phosphonooxy)−6-(2,6,8-trioxo-1,2,6,7,8,9-hexahydro-3H-purin-3-yl)tetrahydro-2H-pyran-2-yl)methyl (R)−6-(((2R,3R,5R,6S)−3,5-dihydroxy-6-methyltetrahydro-2H-pyran-2-yl)oxy)heptanoate | Previously inferred via tandem mass spectrometry (*Artyukhin et al., 2018*; *Curtis et al., 2020*) |  |
| 33 | | 2-Aminobenzoic acid | Commercial product (Sigma-Aldrich) |  |

*Continued on next page*

*Appendix 1—table 9 continued*

| Compound number | SMID ID | IUPAC Name | Evidence | Structure |
|---|---|---|---|---|
| 34 | iglu#3 | ((2R,3S,4S,5R,6R)−3,4,5-trihydroxy-6-(1H-indol-1-yl)tetrahydro-2H-pyran-2-yl)methyl 2-aminobenzoate | Identified via synthesis (This manuscript) | |
| 35 | icas#2 | (2S,3R,5R,6R)−5-hydroxy-2-methyl-6-(((R)−5-oxohexan-2-yl)oxy)tetrahydro-2H-pyran-3-yl 1H-indole-3-carboxylate | Identified via synthesis (*Dong et al., 2016*) | |
| 36 | icas#6.2 | (2S,3R,5R,6R)−5-hydroxy-6-(((2R,5S)−5-hydroxyhexan-2-yl)oxy)−2-methyltetrahydro-2H-pyran-3-yl 1H-indole-3-carboxylate | Identified via synthesis (*Dong et al., 2016*) | |
| SI 1 | | 2-((tert-butoxycarbonyl)-amino)benzoic acid | Characterized via synthesis (This manuscript) | |
| SI 2 | iglu#5 | ((2R,3S,4S,5R,6R)−3,4,5-trihydroxy-6-(1H-indol-1-yl)tetrahydro-2H-pyran-2-yl)methyl nicotinate | Identified via synthesis (This manuscript) | |

*Continued on next page*

*Appendix 1—table 9 continued*

| Compound number | SMID ID | IUPAC Name | Evidence | Structure |
|---|---|---|---|---|
| SI 3 | iglu#7 | ((2*R*,3*S*,4*S*,5*R*,6*R*)−3,4,5-trihydroxy-6-(1*H*-indol-1-yl)tetrahydro-2*H*-pyran-2-yl)methyl (*E*)−2-methylbut-2-enoate | Identified via synthesis (This manuscript) |  |
| SI 4 | iglu#9 | ((2*R*,3*S*,4*S*,5*R*,6*R*)−3,4,5-trihydroxy-6-(1*H*-indol-1-yl)tetrahydro-2*H*-pyran-2-yl)methyl 1*H*-pyrrole-2-carboxylate | Identified via synthesis (This manuscript) |  |
| SI 5 | angl#3 | ((2*R*,3*S*,4*S*,5*R*,6*S*)−6-((2-aminobenzoyl)oxy)−3,4,5-trihydroxytetrahydro-2*H*-pyran-2-yl)methyl 2-aminobenzoate | Proposed structure based on synthesis of a reference sample for MS (This manuscript) |  |
| SI 6 | tyglu#6 | (2*R*,3*R*,4*S*,5*S*,6*R*)−6-(((2-aminobenzoyl)oxy)methyl)−2-((4-(2-aminoethyl)-phenoxy))−5-hydroxy-4-(phosphonooxy)-tetrahydro-2*H*-pyran-3-yl nicotinate | Proposed structure. Initially described (*O'Donnell et al., 2020*) and further inferred via tandem mass spectrometry (This manuscript) |  |

