## [Decision Letter]

**Acceptance summary:**

The strength of your paper lies in the combination of excellent genetics and analytical chemistry, providing strong evidence for the biosynthesis of nematode signaling molecules in a specialized organelle (the LRO). The implication of many different hydrolases in modular biosynthesis of these molecules is also quite interesting and may be broadly important across animals. Overall, your results shed light on a new area of biosynthesis and may have broad implications in the animal kingdom.

**Decision letter after peer review:**

Thank you for submitting your article "Modular metabolite assembly in *C. elegans* lysosome-related organelles" for consideration by *eLife*. Your article has been reviewed by two peer reviewers, and the evaluation has been overseen by a Senior/Reviewing Editor. One of the two individuals involved in review of your submission has agreed to reveal their identity: Jason M Crawford (Reviewer #2).

The reviewers have discussed the reviews with one another and the Reviewing Editor has drafted this decision to help you prepare a revised submission.

Summary:

This very interesting study examines the metabolic capacity of lysosome related organelles (LROs) in the model metazoan *C. elegans*. While the LROs are generally regarded as having recycling functions (e.g., autophagy, etc), Schroeder, Sternberg and co-workers show that the LROs are hotspots for the biosynthesis of select modular glucoside and ascaroside signaling molecules. While many of the ascarosides have established biological activities, the modular glucosides represent previously unknown nematode metabolites now illuminated for further biological study. Their studies suggest that the targeted biosynthesis pathways co-localize with the LROs to efficiently access substrates derived from LRO catabolic processes. This organelle-specific biosynthesis was supported with knockout studies of enzymes required for LRO formation, knockout studies of select biosynthetic enzymes, and imaging studies of a cholinesterase (CEST)-like enzyme involved in the biosynthesis. The strength of this paper lies in the combination of excellent genetics and analytical chemistry, providing strong evidence for the biosynthesis of nematode signaling molecules in a specialized organelle (the LRO). The implication of many different hydrolases in modular biosynthesis of these molecules is also quite interesting and may be broadly important across animals. Overall, these results shed light on a new area of biosynthesis and may have broad implications in the animal kingdom.

Essential revisions:

1) There is concern with a few leaps in logic. Knockout of specific proteins leads to loss of LROs, which is correlated with loss of compounds. However, even with the strong experiments performed here, this correlation does not imply causation. It is possible that the compounds are synthesized outside LROs, but loss of LROs leads to loss of something needed for synthesis, as one example. As another example, individual *cest* homologs are knocked out, leading to the loss of specific esters from the metabolome. However, direct biochemical experiments that firmly establish the reactions that are taking place are not reported. Therefore, we strongly suggest that you make the resulting claims with more precision, and with an acknowledgment of these limitations.

2) Based on the current version of the manuscript, you would have to be an expert in nematode metabolism to truly appreciate the impact of the current study and how it fits in the current literature. This is a testament to the breadth of the current study. The authors should provide a structures table (separate from the mass table provided) including images of the structures, their names, and references to where the structures were originally characterized (many are from this team). Also, it would be helpful to describe new structures in this table and the level of support provided (i.e., some were validated by synthesis whereas others are predictions based on tandem MS). This would provide an easy visual tool that would allow readers to appreciate the impact of the current study, its context with prior studies, and the level of support for individual metabolites.

---

## [Author Response]

Essential revisions:1) There is concern with a few leaps in logic. Knockout of specific proteins leads to loss of LROs, which is correlated with loss of compounds. However, even with the strong experiments performed here, this correlation does not imply causation. It is possible that the compounds are synthesized outside LROs, but loss of LROs leads to loss of something needed for synthesis, as one example. As another example, individual cest homologs are knocked out, leading to the loss of specific esters from the metabolome. However, direct biochemical experiments that firmly establish the reactions that are taking place are not reported. Therefore, we strongly suggest that you to make the resulting claims with more precision, and with an acknowledgment of these limitations.

We agree and revised the text to make clearer that additional experiments are needed (i) to establish exactly where biosynthesis happens and (ii) to confirm that CEST proteins in fact catalyze the formation of ester and amide bonds in the modular metabolites, as we propose. As we state in our manuscript, attempts at heterologous expression have failed so far; however, we have added new data showing that mutating the serine in the conserved catalytic serine-histidine-glutamate triad of *cest-1.1* abolishes production of the same set of metabolites as in the *cest-1.1*(null) mutant, strengthening the case for the proposed biosynthetic function. We added Patrick J. Hu and Joseph C. Kruempel as authors who generated the *cest-1.1* point mutant.

2) Based on the current version of the manuscript, you would have to be an expert in nematode metabolism to truly appreciate the impact of the current study and how it fits in the current literature. This is a testament to the breadth of the current study. The authors should provide a structures table (separate from the mass table provided) including images of the structures, their names, and references to where the structures were originally characterized (many are from this team). Also, it would be helpful to describe new structures in this table and the level of support provided (i.e., some were validated by synthesis whereas others are predictions based on tandem MS). This would provide an easy visual tool that would allow readers to appreciate the impact of the current study, its context with prior studies, and the level of support for individual metabolites.

This is an excellent suggestion. New Table 9 lists all compounds referred to in the text, shows their structures, SMID ID’s, IUPAC names, and indicates whether the structure has been fully validated by synthesis or NMR, or has been proposed based on e.g. MS/MS.